# Beyond Detection: A Structure-Aware Framework for Scene Text Tracking

**Chenmin Yu**[1]  **Liu Yu**[1]  **Daiqing Wu**[2]  **Gengluo Li**[2]  **Zeyu Chen**[1]  **Yu Zhou**[1]

## Abstract

Modern visual object trackers show impressive results on general targets, yet their performance drops substantially when dealing with scene text. Although currently underexplored, tracking text in videos is essential for dynamic text manipulations such as segmentation, removal, and editing. To fill this gap, this paper formalizes this specific task as Scene Text Tracking and presents the first systematic work for it. We identify three primary challenges in this task: 1) severe geometric distortions from perspective shifts, 2) high visual ambiguity across different instances, and 3) high sensitivity to fine-grained structural details. To address these issues, we propose SymTrack, a unified detection-free framework with synergistic dual-branch design. It integrates a Cross-Expert Calibration mechanism to reduce semantic bias, along with a Predictive Token Rectification mechanism to correct structural imbalances, complemented by an Adaptive Inference Engine that stabilizes predictions under motion constraints. Considering the lack of dedicated benchmarks for this task, we utilize three datasets from video text spotting to construct a benchmark with high-quality annotations. Extensive experiments demonstrate that SymTrack sets the new state-of-the-art on all three benchmarks, outperforming previous best trackers by up to 11.97% AUC on BOVText$_{SOT}$. Overall, our work promotes efficient and thorough text tracking, paving the way toward more generalized video text manipulation. Code is available at https://github.com/EdisonYCM/SymTrack.

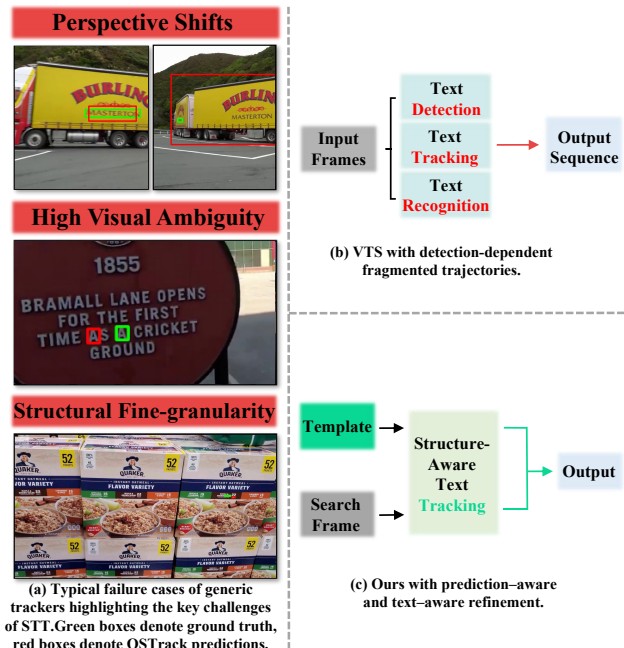

(a) Typical failure cases of generic trackers highlighting the key challenges of STT. Green boxes denote ground truth, red boxes denote OSTrack predictions.

(b) VTS with detection-dependent fragmented trajectories.

(c) Ours with prediction–aware and text–aware refinement.

*Figure 1.* Motivation and paradigm comparison. (a) Typical failure cases of scene text tracking. From top to bottom: Perspective shifts leading to drastic shape distortion; high visual ambiguity from nearby similar texts; fine-grained sensitivity in densely packed small texts. (b) VTS treats tracking as a by-product, where detection failures fragment text trajectories. (c) We propose a detection-free, unified framework that directly models text structure.

## 1. Introduction

Text instances in videos serve as a crucial high level semantic medium, offering rich information for scene understanding, human-computer interaction, and assistive technologies (Jaderberg et al., 2016; Singh et al., 2019; Cheng et al., 2019). In recent years, scene text research has made substantial progress across a broad range of tasks, including detection (Wang et al., 2022; Chen et al., 2021b), recognition (Qiao et al., 2020; Yang et al., 2025a), spotting (Lyu et al., 2025; Wei et al., 2022), processing (Shu et al., 2025; Zeng et al., 2024; Chen et al., 2026), and understanding (Shu et al., 2026; Zeng et al., 2023). By comparison, scene text in videos has received relatively less systematic attention, although recent studies have begun to explore video text understanding and reasoning from spatio-temporal or

---

[1]VCIP & TMCC & DISSec, College of Computer Science & College of Cryptology and Cyber Science, Nankai University [2]Institute of Information Engineering, Chinese Academy of Sciences, Beijing, China. Correspondence to: Yu ZHOU <yzhou@nankai.edu.cn>.

*Proceedings of the 43$^{rd}$ International Conference on Machine Learning*, Seoul, South Korea. PMLR 306, 2026. Copyright 2026 by the author(s).

instance-oriented perspectives (Zhang et al., 2025a; Yang et al., 2025b; Zhang et al., 2025b). Currently, the dominant paradigm for processing video text is Video Text Spotting (VTS), which aims to simultaneously detect, track, and recognize all text instances (Wang et al., 2017; Cheng et al., 2020; Wu et al., 2025; 2024a; Yin et al., 2016). This end-to-end approach provides a comprehensive semantic understanding of the video content. However, VTS frameworks are often computationally intensive, requiring per-frame detection and recognition. Furthermore, their reliance on detection can lead to fragmented trajectories when instances are occluded or blurred (Cheng et al., 2020; 2019; Lei et al., 2025). This creates a need for a more efficient and flexible alternative: Scene Text Tracking (STT), which focuses on the dedicated task of localizing a specific text instance throughout a video. Inspired by visual object tracking (Fan et al., 2019; Huang et al., 2021), we define the STT task as predicting the precise location of an arbitrary text target in subsequent frames, given only its initial state in the first frame. STT transforms paradigm from frame-oriented to instance-oriented, improving tracking reliability while avoiding redundant computations and fragmented trajectories, showing potential for generalized video text manipulation. For training and evaluation, the initial target bounding box is either provided by the dataset or manually specified.

Despite its potential, STT remains a largely underexplored task with no dedicated methods. Conceptually, this task can be approached via two existing paradigms. The first is to apply generic visual object trackers, which feed both template and search frames into a visual encoder for joint feature learning and relation modeling (Ye et al., 2022; Chen et al., 2022; Cai et al., 2024; Zheng et al., 2024; Lin et al., 2024). The second is to adapt VTS methods via external cascading, as illustrated in Figure 1(b), where text instances are first detected separately and then associated across frames (Cheng et al., 2019; 2020; Wu et al., 2025; 2024a).

However, these paradigms do not transfer well when applied directly. We attribute this limitation to three core challenges specific to scene text, as outlined in Figure 1(a): **1). Feature Degradation from Perspective Shift.** Severe distortions from perspective shifts alter the geometric structure of planar text instances. In generic trackers, this leads to a misalignment between the template and search region features. Text identification relies on high-frequency structural features, but this misalignment creates an information bottleneck, as the shallow head struggles to extract the target from a noisy, high-entropy feature map. In VTS pipelines, these distortions can cause detection failures, leading to lost tracks. **2). High Visual Ambiguity.** Text instances suffer from similar character structures, acting as powerful distractors. This is a critical weakness for generic models, which lack text-specific feature modeling and easily drift to nearby, similar-looking text (Gao et al., 2022; Cai et al.,

2023). **3). Structural Fine-granularity.** Text demands high sensitivity to fine-grained structural details, where minor localization drift can alter the perceived content and lead to tracking failure. This is aggravated by limited temporal modeling in predominant frame-pair matching approaches.

To address these challenges, we advocate for a paradigm shift and propose SymTrack, a detection-free framework grounded in a synergistic modeling architecture, overviewed in Figure 1(c). The encoder features a dual-branch design to acquire both spatio-temporal representations and text-specific features. In its first branch, the Predictive Token Rectification (PTR) module receives features fused with spatial and temporal clues from the visual backbone, and preemptively refines the feature space to resolve the information bottleneck, countering the challenge of perspective shift. In parallel, the second branch employs a textual feature expert and a Cross-Expert Calibration (CEC) mechanism to provide textual priors, effectively resolving high visual ambiguity. This dual-branch design provides the prediction head with a feature representation that is both spatially robust and semantically precise. Finally, in the prediction head, we propose the Adaptive Inference Engine (AIE) module, which stabilizes predictions by adaptively regulating the model's search region and suppressing temporal jitter, enhancing sensitivity to fine-grained features.

Our main contributions are summarized as follows:

- We provide a systematic analysis of STT, identifying core challenges including severe distortions from perspective shifts, high visual ambiguity across instances, and fine-grained structural sensitivity.

- We propose SymTrack, a unified architecture equipped with PTR, CEC and AIE, which respectively alleviate structural imbalance, semantic bias, and motion limitation problems of existing trackers.

- Considering the lack of dedicated STT benchmarks, we build upon three datasets from VTS and ensure high-quality annotations. On these benchmarks, our proposed SymTrack sets the new state-of-the-art performance.

## 2. Related Works

Our work establishes a dedicated paradigm for STT, distinct from the two dominant philosophies in the tracking literature: (i) generic visual object tracking and (ii) tracking-by-detection family, which also forms the basis of most VTS systems. We review the architectural evolution in both domains and highlight the methodological gaps that motivate the need for dedicated STT frameworks.

## 2.1. Generic Visual Object Tracking

**Spatial Modeling and the Information Bottleneck.** The dominant tracking paradigm has shifted from lightweight Siamese matching schemes toward powerful, relation-modeling architectures. Early Siamese trackers emphasized efficient template–search matching and robust online inference, but struggled with large appearance changes and complex backgrounds (Li et al., 2019). Transformer-based trackers like TransT (Chen et al., 2021a) and STARK (Yan et al., 2021) introduced dedicated modules for explicit template-search feature interaction. This trend culminated in one-stream architectures (Ye et al., 2022; Chen et al., 2022; Cui et al., 2022; Xie et al., 2022) that perform joint feature extraction and relation modeling within a single backbone (Dosovitskiy et al., 2021). However, to maintain real-time efficiency, these powerful backbones are typically paired with shallow prediction heads (Danelljan et al., 2019; Bhat et al., 2019). This design, though effective for generic objects, imposes a structural imbalance on scene text, forming an information bottleneck that weakens fine-grained sensitivity and discriminative robustness, leading to localization drift and confusion with distractors.

**Temporal Modeling from Static Pairs to Dynamics.** Early trackers were largely temporally agnostic, treating tracking as a series of independent frame-pair matching problems. To address this, some works introduced online update mechanisms to adapt the template over time, either through learnable updaters (Chen et al., 2020; Guo et al., 2022; Zhang et al., 2019) or dynamic reference selection strategies (Yan et al., 2021; Danelljan et al., 2019). Recently, temporal modeling has advanced further through explicit feature propagation, such as ODTrack (Zheng et al., 2024), which encodes target features as token sequences, and autoregressive approaches like ARTrack (Wei et al., 2023) and SeqTrack (Chen et al., 2023), which sequentially predict trajectories. Despite these advances, such methods are trained on generic object datasets (Fan et al., 2019; Huang et al., 2019; Lin et al., 2014; Liu et al., 2026) and thus learn only coarse motion priors. These priors are inadequate for STT: they fail to handle the feature mismatches caused by perspective shifts and the fine-grained sensitivity required for text localization. Moreover, information propagation can amplify small localization errors over time, leading to drift and eventual tracking failure. This reveals a persistent gap, as spatial refinement and temporal modeling remain decoupled, neither tailored to the unique properties of scene text.

## 2.2. Tracking-by-Detection Paradigm

An alternative family treats tracking predominantly as per-frame detection followed by association, which is prevalent in multi-step systems (Bewley et al., 2016; Wojke et al.,

2017; Sun et al., 2021). In its workflow, an object detector is applied on a per-frame basis to generate candidate bounding boxes. Subsequently, a downstream association module often involving heuristics like embedding matching or simple appearance similarity, is used to associate these independent detections into trajectories (Bochinski et al., 2017; Cao et al., 2023; Zhang et al., 2022). Tracking-by-detection has been widely used in multi-object (Zhang et al., 2021; Chu et al., 2023) and specialized pipelines because it decouples localization from association and leverages strong detectors.

**Inherent Limitations.** The tracking-by-detection paradigm is fundamentally constrained by the per-frame detector's accuracy (Zhang et al., 2022), a limitation magnified in video text scenarios. Text detectors often fail under perspective-induced distortions or dense, visually ambiguous instances (Wu et al., 2024a), where a single missed or false detection can break temporal continuity and cause unrecoverable identity loss (Bergmann et al., 2019). Moreover, redundant frame-wise detection is both computationally inefficient and conceptually misaligned with tracking's emphasis on temporal coherence. These methods relegate tracking to a downstream association task. In contrast, our work advocates a tracking-first, detection-free philosophy centered on continuous temporal modeling.

## 3. Method

In this section, we introduce the proposed SymTrack framework. We follow the tracker's operational workflow to detail its components, illustrating its synergistic dual-branch design for feature representation. After feature calibration, the refined map is fed into the prediction head for target localization, where the AIE further stabilizes the output during inference, as shown in Figure 2.

### 3.1. Predictive Token Rectification

SymTrack's workflow begins with the template image patch and the current search region patch. They are first splitted and flattened into sequences of patches, which are then projected into patch embeddings. Learnable position embeddings are added to produce the template tokens $\mathcal{Z}$ and search region tokens $X_t$. These tokens are fed into a ViT backbone (Dosovitskiy et al., 2021), which performs joint feature extraction and relation modeling to produce an initial search feature map $F_x$. As identified in Section 1, this raw feature map $F_x$ still suffers from the information bottleneck, forcing a shallow prediction head to disambiguate the target from noisy features.

To address this, we introduce the Predictive Token Rectification (PTR) module as the first branch of our encoder. The PTR module is a lightweight, trainable network that preemptively refines the search feature landscape using high-level

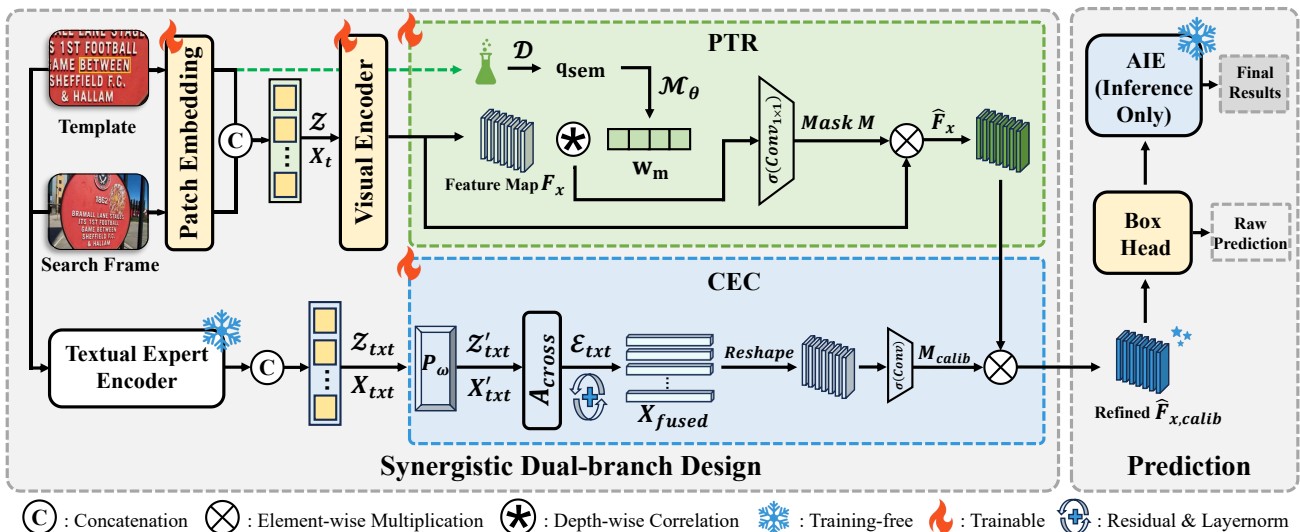

*Figure 2.* Overview of the SymTrack framework, which employs a synergistic dual-branch design. The top branch performs Predictive Token Rectification (PTR). It distills a semantic query $\mathbf{q}_{\text{sem}}$ from the template tokens $\mathcal{Z}$ to generate a modulation mask $M$. This mask is applied to the main visual feature map $F_x$ to produce a rectified map $\hat{F}_x$, addressing structural imbalances. The parallel branch performs Cross-Expert Calibration (CEC). It uses a textual feature encoder to generate a text-specific calibration mask $M_{calib}$ that resolves high visual ambiguity. The outputs from PTR and CEC are fused via element-wise multiplication into a refined map $\hat{F}_{x,calib}$. This map is fed to the prediction head, where the training-free Adaptive Inference Engine (AIE) stabilizes the final output during inference.

semantics extracted from the template. First, the holistic appearance of the target is distilled from the template tokens $\mathcal{Z}$ into a potent semantic query vector $\mathbf{q}_{\text{sem}}$:

$$\mathbf{q}_{\text{sem}} = \mathcal{D}(\mathcal{Z}) = \frac{1}{N_z} \sum_{i=1}^{N_z} z_i. \tag{1}$$

This semantic query $\mathbf{q}_{\text{sem}}$ is then projected by a mapping network $\mathcal{M}_\theta$, implemented as a multi-layer perceptron, to generate a channel-wise modulation field $\mathbf{w}_m \in \mathbb{R}^C$.

$$\mathbf{w}_m = \mathcal{M}_\theta(\mathbf{q}_{\text{sem}}). \tag{2}$$

This field is used to generate a probabilistic gating mask $M \in [0,1]^{H \times W}$ via a function $\mathcal{G}_\phi$. This function models the spatial correlation between the search features $F_x$ and the modulation field $\mathbf{w}_m$:

$$M = \sigma\big(\mathcal{G}_\phi(F_x, \mathbf{w}_m)\big) = \sigma\big(\text{Conv}_{1\times1}(F_x \circledast \mathbf{w}_m)\big). \tag{3}$$

Here, $\circledast$ represents a depth-wise correlation, efficiently implemented as a depth-wise convolution where $\mathbf{w}_m$ is injected as a channel-wise bias. The final rectified feature map $\hat{F}_x$ is obtained by applying this probabilistic gate via element-wise multiplication ($\odot$):

$$\hat{F}_x = F_x \odot M. \tag{4}$$

This rectification process is crucial for STT. Rather than correcting geometric distortions, which can introduce resampling noise, PTR performs feature-level rectification, easing the information bottleneck and mitigating issues caused by perspective shifts and fine-grained structural sensitivity.

## 3.2. Cross-Expert Calibration

While the PTR branch refines the spatio-temporal feature landscape, it does not resolve the high visual ambiguity inherent in video text. Generic visual features fail to distinguish the target from similar distractors. Therefore, we introduce a parallel branch to provide text-specific guidance.

In parallel to the main visual backbone, the template and search frames are also processed by a frozen, high-resolution backbone pre-trained on large-scale text-centric data. In this work, we adopt the visual encoder of TokenFD (Guan et al., 2025). This branch extracts high-fidelity, text-specific feature sets for the template $\mathcal{Z}_{txt} \in \mathbb{R}^{N_{zt} \times C_{txt}}$ and the search region $\mathcal{X}_{txt} \in \mathbb{R}^{N_{zt} \times C_{txt}}$. These specialized features are then fed into the Cross-Expert Calibration (CEC) module to guide the main tracking branch. First, the high-dimensional textual features are projected into a lower-dimensional embedding space via a shared linear layer $\mathcal{P}_\omega$:

$$\mathcal{Z}'_{txt} = \mathcal{P}_\omega(\mathcal{Z}_{txt}), \quad \mathcal{X}'_{txt} = \mathcal{P}_\omega(\mathcal{X}_{txt}). \tag{5}$$

We then employ a multi-head cross-attention mechanism, $\mathcal{A}_{cross}$, to enhance the search region features with template-specific textual information, where the projected search features serve as the query:

$$\mathcal{E}_{txt} = \mathcal{A}_{cross}(\mathcal{X}'_{txt}, \mathcal{Z}'_{txt}, \mathcal{Z}'_{txt}). \tag{6}$$

The enhanced representation is obtained via a residual con-

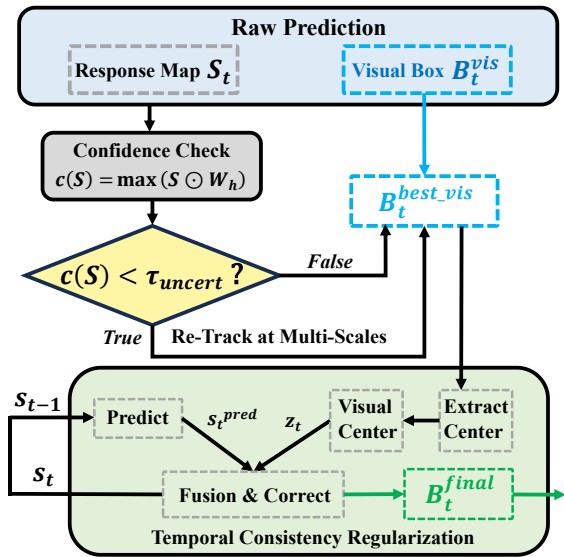

*Figure 3.* The structure of AIE, which uses confidence for adaptive search and a module for temporal regularization.

nection and layer normalization:

$$\mathcal{X}_{fused} = \text{LayerNorm}(\mathcal{X}'_{txt} + \mathcal{E}_{txt}). \quad (7)$$

Finally, the fused token sequence $\mathcal{X}_{fused}$ is reshaped and passed through a lightweight convolutional head $\mathcal{H}_\psi$ to generate a spatial calibration mask $M_{calib}$ with values $\in [0, 1]$:

$$M_{calib} = \sigma(\mathcal{H}_\psi(\text{Reshape}(\mathcal{X}_{fused}))). \quad (8)$$

This calibration mask injects crucial textual priors into the main pipeline, forcing the model to suppress background distractors and thus effectively resolving the high visual ambiguity challenge.

### 3.3. Adaptive Inference Engine

The outputs from both branches are now ready for final fusion. The spatially rectified feature map $\hat{F}_x$ (from PTR) is modulated by the textually calibrated mask $M_{calib}$ (from CEC) via element-wise multiplication ($\odot$):

$$\hat{F}_{x,calib} = \hat{F}_x \odot M_{calib}. \quad (9)$$

This step forces the tracker to focus on candidates that are coherent in both spatio-temporal and textual feature spaces. This final, fully calibrated feature map $\hat{F}_{x,calib}$ is then fed into a lightweight prediction head to generate the target response map.

During inference, we introduce a training-free module to ensure high sensitivity to fine-grained features: the Adaptive Inference Engine (AIE). As shown in Figure 3, the AIE dynamically modulates the tracker's output to handle observation uncertainty in two ways. First, it adjusts the search

region dynamically to handle geometric distortions caused by perspective shifts. The tracker's predictive confidence $c(S)$ is quantified by the peak value of the score map after applying a Hann window. If this confidence $c(S)$ falls below a threshold $\tau_{\text{uncert}}$, the model re-runs inference on search regions with alternative, pre-defined scale factors $\mathcal{S}_{factors}$, achieving multi-scale robustness only when needed.

Second, to enforce a smooth trajectory and address the challenge of fine-grained sensitivity, we employ a temporal regularization model. We formulate tracking within a linear state-space model, where the target's kinematic state $s_t = [c_x, c_y, v_x, v_y]^T$ evolves according to a constant velocity prior. The model fuses the motion prediction with the visual tracker's output $z_t$ using a fusion weight $\alpha_{kalman}$ to yield a filtered posterior estimate $s_t$. By exploiting motion continuity through temporal regularization, the AIE effectively mitigates jitter and long-term drift, stabilizing localization for fast-moving or scrolling text.

## 4. Experiments

### 4.1. Implementation Details

**Datasets.** Since STT is an underexplored task, no dedicated benchmarks are readily available. However, existing Video Text Spotting (VTS) benchmarks (He et al., 2025; Wu et al., 2024b; 2025) provide high-fidelity instance IDs and dense polygonal annotations, which enables their conversion into the Single Object Tracking (SOT) format through a systematic pipeline of trajectory grouping, continuity splitting, and format conversion. Our conversion method is designed to minimize annotation noise, ensuring that the resulting STT benchmarks largely preserve both the annotation precision and the high data quality of their VTS origins. **First**, for each source video, we parse all annotations and group them by their unique instance ID. **Second**, we perform continuity splitting. We sort the annotations for each instance by frame ID and iterate through the sequence, splitting the trajectory into a new, separate SOT sample whenever an annotation for that instance is entirely absent for one or more frames. This process is vital, as VTS annotations often handle a target's complete disappearance by dropping the annotation. Crucially, this method retains all frames where the target is partially occluded, as long as a partial annotation is still provided, ensuring our benchmark retains challenging occlusion scenarios. **Next**, we filter out and discard all resulting tracklets shorter than a minimum threshold of five frames. **Finally**, for SOT format conversion, we generate a self-contained SOT sample for each valid tracklet, converting polygon annotations into minimal enclosing bounding boxes and writing them to the corresponding ground truth file. To avoid data duplication, we use symbolic links to create a re-indexed image sequence. This procedure yields our three new SOT-formatted benchmarks:

*Table 1.* State-of-the-art comparison on ArTVideo$_{SOT}$, DSText$_{SOT}$, and BOVText$_{SOT}$. Subscripts denote the corresponding configuration and input resolution. "V-L" denotes visual-language tracking. VTS models are excluded on BOVText$_{SOT}$ due to the lack of Chinese character sets in their public implementations. The top two results are highlighted with **bold** and underlined fonts, respectively.

| Type | Method | Venue | ArTVideo$_{SOT}$ | | | DSText$_{SOT}$ | | | BOVText$_{SOT}$ | | |
|---|---|---|---|---|---|---|---|---|---|---|---|
| | | | AUC (%) | P$_{Norm}$ (%) | P (%) | AUC (%) | P$_{Norm}$ (%) | P (%) | AUC (%) | P$_{Norm}$ (%) | P (%) |
| Vision-only | SiamRPN++ (Li et al., 2019) | CVPR2019 | 56.40 | 67.30 | 71.90 | 44.40 | 54.40 | 63.20 | 58.70 | 71.50 | 65.50 |
| | STARK (Yan et al., 2021) | ICCV2021 | 70.37 | 83.48 | 86.84 | 57.59 | 68.63 | 78.01 | 61.92 | 75.16 | 76.33 |
| | OSTrack$_{256}$ (Ye et al., 2022) | ECCV2022 | 64.86 | 77.95 | 81.99 | 52.50 | 63.49 | 70.83 | 58.67 | 73.04 | 74.03 |
| | OSTrack$_{384}$ (Ye et al., 2022) | ECCV2022 | 64.80 | 77.82 | 82.05 | 54.83 | 66.51 | 74.44 | 59.18 | 72.68 | 73.30 |
| | AiATrack (Gao et al., 2022) | ECCV2022 | 66.41 | 77.96 | 81.77 | 57.92 | 68.12 | 79.02 | 64.16 | 75.07 | 73.42 |
| | SeqTrack$_{L384}$ (Chen et al., 2023) | CVPR2023 | 64.35 | 76.46 | 80.92 | 54.63 | 65.81 | 74.19 | 60.42 | 76.18 | 76.70 |
| | ARTrack$_{256}$ (Wei et al., 2023) | CVPR2023 | 64.85 | 78.81 | 79.53 | 48.53 | 56.12 | 65.20 | 62.75 | 72.28 | 73.01 |
| | GRM$_{256}$ (Gao et al., 2023) | CVPR2023 | 68.22 | 79.84 | 83.30 | 53.05 | 63.87 | 71.04 | 59.59 | 72.12 | 72.82 |
| | GRM$_{384}$ (Gao et al., 2023) | CVPR2023 | 68.47 | 80.65 | 83.64 | 55.51 | 66.07 | 74.63 | 59.13 | 71.02 | 71.66 |
| | ROMTrack (Cai et al., 2023) | ICCV2023 | 70.62 | 83.32 | 87.13 | 56.82 | 68.79 | 75.61 | 62.82 | 73.74 | 74.90 |
| | ODTrack (Zheng et al., 2024) | AAAI2024 | 69.81 | 83.54 | 86.68 | 62.71 | 75.84 | 84.26 | 64.74 | 77.74 | 78.45 |
| | **SymTrack (Ours)** | ICML2026 | **77.74** | **91.29** | **95.88** | **70.66** | **83.61** | **91.83** | **77.06** | **90.05** | **90.18** |
| V-L | DUTrack$_{256}$ (Li et al., 2025b) | CVPR2025 | 68.73 | 82.46 | 86.87 | 60.57 | 72.77 | 81.31 | 65.09 | 78.98 | 79.04 |
| | DUTrack$_{384}$ (Li et al., 2025b) | CVPR2025 | 72.09 | 85.97 | 89.36 | 63.63 | 76.72 | 85.00 | 65.08 | 79.41 | 79.30 |
| VTS | TransVTSpotter (Wu et al., 2025) | NeurIPS2021 | 8.84 | 78.11 | 38.07 | 4.93 | 75.21 | 67.80 | - | - | - |
| | TransDETR (Wu et al., 2024a) | IJCV2024 | 9.18 | 78.75 | 43.31 | 5.08 | 76.09 | 69.79 | - | - | - |

*Table 2.* Statistics of the curated STT datasets. We detail the number of videos, original VTS instances, and final generated SOT tracklets for both training and testing splits.

| Dataset | Split | Videos | IDs (VTS) | Tracklets (SOT) |
|---|---|---|---|---|
| ArTVideo$_{SOT}$ | Train | 40 | 1202 | 1183 |
| | Test | 20 | 246 | 239 |
| DSText$_{SOT}$ | Train | 90 | 48733 | 15708 |
| | Test | 50 | 15041 | 11101 |
| BOVText$_{SOT}$ | Train | 1541 | 66668 | 52290 |
| | Test | 480 | 20085 | 16380 |

*Table 3.* Comparison of model accuracy, parameters, and inference speed. Results are evaluated on ArTVideo$_{SOT}$.

| Method | AUC (%) | Params (M) | Speed (fps) |
|---|---|---|---|
| SymTrack (Ours) | 77.74 | 395.917 | 22.07 |
| SymTrack *w/o* TokenFD | 75.45 | 92.674 | 89.02 |
| SeqTrack | 64.35 | 306.525 | 15.88 |

**ArTVideo$_{SOT}$**, **DSText$_{SOT}$**, and **BOVText$_{SOT}$**, evaluated using metrics AUC, P$_{Norm}$, and P. The statistics of the datasets are detailed in Table 2.

**Training and Optimization.** Our model is trained in two stages. **Stage 1 (General Pre-training):** Following standard practice (Ye et al., 2022), we pre-train the main tracking branch, which comprises its visual backbone (a ViT-B (Dosovitskiy et al., 2021) initialized with MAE (He et al., 2022) weights) and the PTR module. The model is trained on the combined training splits of LaSOT (Fan et al., 2019), GOT-10K (Huang et al., 2019), COCO (Lin et al., 2014), and TrackingNet (Wang et al., 2020) to build a robust foundational model for spatio-temporal modeling. This stage equips the model to handle complex dynamics and distortions, which is a prerequisite for the subsequent text-specific fine-tuning. We adopt AdamW (Loshchilov & Hutter, 2019) as the optimizer and train for 300 epochs with 30k samples per epoch. The initial learning rates are set to $1e-4$ for the backbone and PTR, and $1e-5$ for the rest, with a weight decay of 0.1. The learning rate is reduced by a factor of 10 after 240 epochs. Training is conducted on four NVIDIA A6000 GPUs with a total batch size of 32. **Stage 2 (Text Fine-tuning):** We freeze the textual feature expert, inspired by the vision transformer design of TokenFD (Guan et al.,

2025), and fine-tune the remaining trainable components on our curated text-tracking datasets. Learning rates are set to $5e-5$ for ViT-B and $5e-4$ for the rest. This stage is performed on four NVIDIA RTX 4090 GPUs with a total batch size of 16.

**Inference.** During inference, we employ the AIE and set the confidence threshold $\tau_{uncert}$ to 0.98, the alternative scale factors $\mathcal{S}_{factors}$ to $\{0.95, 1.05\}$, and the fusion weight $\alpha_{kalman}$ to 0.5. Further, we conduct comparative experiments in model parameters and inference speed, as shown in Table 3. More discussion of these hyperparameters and computational costs is provided in the appendix.

## 4.2. Comparison with State-of-the-Art Methods

**ArTVideo$_{SOT}$.** Derived from ArTVideo (He et al., 2025), this benchmark features artistic text with significant appearance variations. As shown in Table 1, SymTrack achieves a top performance of **77.74%** AUC. This represents a significant lead of **+5.65%** AUC over the best vision-language method DUTrack$_{384}$ (Li et al., 2025b) and **+7.12%** AUC over the strongest vision-only tracker ROMTrack (Cai et al., 2023). It highlights the robustness of our PTR module in handling the severe perspective shifts common in artistic text, while the AIE ensures stable localization against fine-grained structural changes.

**DSText$_{SOT}$.** Adapted from DSText (Wu et al., 2024b), this

*Table 4.* Performance of fine-tuned ODTrack (Zheng et al., 2024) on the proposed scene text tracking datasets.

| Datasets | AUC (%) | $P_{Norm}$ (%) | $P$ (%) |
|---|---|---|---|
| ArTVideo$_{SOT}$ | 71.83 | 84.89 | 90.94 |
| DSText$_{SOT}$ | 65.61 | 77.73 | 87.41 |
| BOVText$_{SOT}$ | 67.23 | 79.92 | 80.10 |

*Table 5.* Ablation studies on CEC, AIE and PTR. Experiments are conducted on ArTVideo$_{SOT}$.

| # | CEC | AIE | PTR | AUC (%) | $P_{Norm}$ (%) | $P$ (%) |
|---|---|---|---|---|---|---|
| 1 | ✗ | ✗ | ✗ | 69.50 | 83.42 | 87.02 |
| 2 | ✗ | ✔ | ✗ | 70.85 | 84.30 | 87.23 |
| 3 | ✗ | ✗ | ✔ | 74.46 | 88.37 | 91.66 |
| 4 | ✗ | ✔ | ✔ | 75.45 | 89.31 | 93.55 |
| 5 | ✔ | ✗ | ✔ | 76.58 | 90.21 | 93.97 |
| **6** | ✔ | ✔ | ✔ | **77.74** | **91.29** | **95.88** |

benchmark focuses on densely arranged text instances. Sym-Track again sets a new state-of-the-art, which represents a significant gain of **7.03%** AUC over DUTrack$_{384}$ and **7.95%** AUC over the best vision-only tracker ODTrack (Zheng et al., 2024), validating our model's ability to resolve high visual ambiguity. This is primarily due to the CEC mechanism providing text-specific priors, synergizing with the PTR module's feature rectification to distinguish the target from dense distractors.

**BOVText$_{SOT}$.** This benchmark is created from BOV-Text (Wu et al., 2025), which contains overlaid text with transparency and complex backgrounds. SymTrack achieves **77.06%** AUC, establishing a substantial lead of **11.97%** AUC over DUTrack$_{256}$ and **12.32%** AUC over ODTrack. These improvements highlight the robustness of our framework's CEC in disentangling the target from complex backgrounds, while PTR maintains precise localization on low-contrast, fine-grained structures.

**Comparisons with VTS Tracking Performance.** To validate our "tracking without detection" philosophy, we compare SymTrack against end-to-end VTS methods (Wu et al., 2024a; 2025) on ArTVideo$_{SOT}$ and DSText$_{SOT}$, converting their results to the SOT format. The performance gap shown in Table 1 is stark. On DSText$_{SOT}$, our SymTrack outperforms TransDETR by **+65.58%** AUC, and on ArTVideo$_{SOT}$, the margin is even larger at **+68.56%** AUC, confirming a near-total failure of VTS models in continuous tracking. While their P$_{Norm}$ scores are reasonable, SymTrack still surpasses TransDETR by **+12.54%** P$_{Norm}$ and **+52.57%** Precision on ArTVideo$_{SOT}$, and by **+7.52%** P$_{Norm}$ and **+22.04%** Precision on DSText$_{SOT}$. This gap highlights a fundamental flaw: VTS models reliant on per-frame detection break continuity on a single missed detection, leading to catastrophic failure under metrics. This demonstrates that a detection-free framework with continuous temporal modeling and calibrated text feature representations outperforms end-to-end pipelines combining detection, tracking, and recognition, highlighting the advantages of our design for STT. Further discussion and an additional ID-agnostic evaluation of VTS baselines are provided in Section C.3.

**Fine-tuned Baseline on Scene Text Tracking.**

A potential concern with our main comparison in Table 1 is that baseline trackers are evaluated in a zero-shot manner, having never been trained on scene text. To isolate the

impact of architecture versus data, we conduct a supplementary experiment. We select the strongest vision-only baseline, ODTrack, for this test. We opt against fine-tuning DUTrack, as it leverages a large language model to generate dynamic descriptions based on object category, which is fundamentally different from our visual STT task. Moreover, modifying its complex large language model component is non-trivial for this domain. Therefore, to ensure a fair architectural comparison, we fine-tune ODTrack on the training splits of our curated ArTVideo$_{SOT}$, DSText$_{SOT}$, and BOVText$_{SOT}$ datasets. The results, presented in Table 4, show that while fine-tuning provides a modest performance lift, the fine-tuned ODTrack remains significantly outperformed by our SymTrack. For instance, on BOVText$_{SOT}$, our method maintains a substantial **+9.83%** AUC margin over the fine-tuned ODTrack. This crucial result demonstrates that the performance gap is not merely a data-domain issue but a fundamental architectural limitation. Generic trackers, even when adapted to the data, lack the specialized components to overcome the core challenges of scene text. Their inherent structural imbalances and absence of text-specific priors render them ill-equipped to handle the severe distortions, high visual ambiguity, and fine-grained sensitivity of text instances. This validates the necessity of our dedicated architecture for robust STT.

### 4.3. Analytical Experiments

**The Importance of CEC, AIE and PTR.** Table 5 demonstrates the strong synergistic effect of our core designs. As shown in Table 5, by integrating the PTR module (#3), the performance increases by **+4.96%** AUC, demonstrating its crucial role in resolving the information bottleneck between the backbone and the shallow prediction head. Building upon this, the inclusion of our CEC mechanism (#5) further boosts the AUC to 76.58% (**+2.12%**), which validates that our dual-branch design successfully counters the semantic bias of generic trackers. Finally, enabling the training-free AIE (#6) on the full model adds another **+1.16%** AUC, confirming its effectiveness in ensuring high sensitivity to fine-grained feature during inference.

**Comparison of Response and Attention Maps with and without CEC.** In Figure 4, to exclude the influence of intermediate tracked frames, we use the model in rows 4

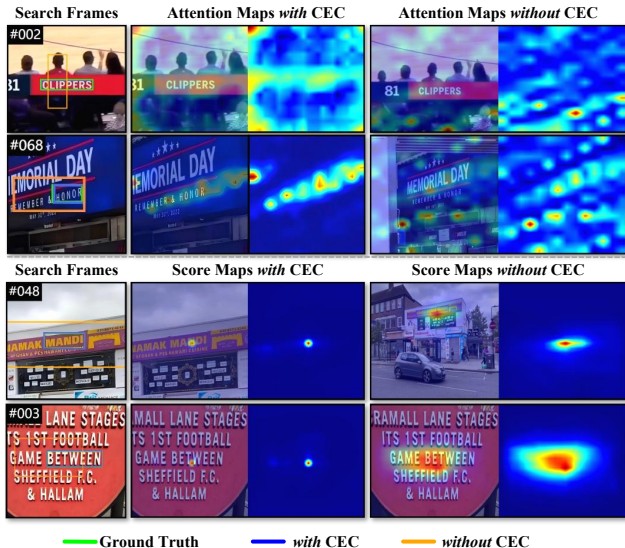

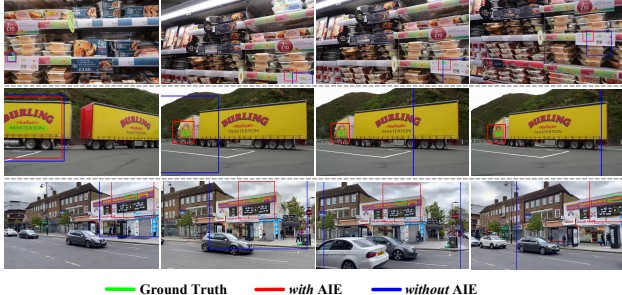

*Figure 5.* Visualization comparison of search region with and without AIE. Zoom in for better view.

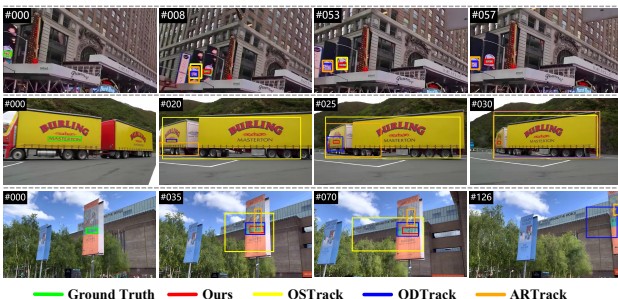

*Figure 6.* Qualitative comparison results of SymTrack with other SOTA trackers on three challenging sequences from ArTVideo$_{SOT}$ and DSText$_{SOT}$ benchmark. Better viewed in color with zoom-in.

*Figure 4.* Visualization comparison of search frames attention maps and score maps with and without CEC. Zoom in for better view.

*Table 6.* Ablation studies on AIE. Results are evaluated on DSText$_{SOT}$.

| Model Variants | Avg. SRC (%) | Avg. IoU (%) |
|---|---|---|
| **SymTrack** | **95.25** | **71.96** |
| *w/o* AIE | 83.27 | 69.54 |

and 6 of Table 5 for comparison, where CEC is removed to isolate the effects of text-specific guidance. We first analyze the attention maps of the search region from the last block of the transformer encoder. The first column shows the zoomed-in search frame, while the second and third columns present the visualization results and the corresponding raw attention maps. As shown in Figure 4, when facing challenging scenarios such as occlusion, dense text interference, and viewpoint changes, CEC effectively suppresses background activation and enhances the perception of target text boundaries. Then, we compare the score map of the tracker input prediction head. Similarly, we adopt a three-column visualization layout. The score map represents the spatial confidence distribution predicted by the prediction head, where each location indicates the model's confidence that the target center lies at the corresponding position within the search region. In Figure 4, the response map without CEC exhibits dispersed or ambiguous peaks, reflecting degraded localization confidence caused by feature misalignment and representation imbalance between the backbone and the prediction head. In contrast, with our proposed CEC, the response map shows sharp and well-localized peaks, achieving robust and accurate STT with strong resilience to interference and generalization across diverse scenarios.

**Impact of Adaptive Inference Engine.** We analyze the effectiveness of our training-free AIE in handling the severe geometric transformations unique to scene text. Generic trackers often fail when text undergoes rapid changes in scale and aspect ratio due to perspective shifts, causing the target to fall outside their static or slowly-adapting search areas. Our AIE is designed to mitigate this by dynamically controlling the search region and regularizing the trajectory. To quantify this, we introduce the Search Region Coverage (SRC) metric, defined as $\text{Area}(B_{search} \cap B_{gt})/\text{Area}(B_{gt})$, which measures how well the search area manages to contain the ground truth target. As shown in Table 6, our full Sym-Track model with AIE achieves a significant improvement of **+11.98%** over the baseline without AIE. This demonstrates that our AIE proactively adapts the search region to keep the fast-moving, deforming text target within view. This superior coverage directly translates to improved tracking accuracy, with the average IoU increasing from 69.54% to 71.96%. The qualitative results in Figure 5 further validate this. In challenging scenarios involving perspective distortion, tracker without AIE fails to adapt and loses the target. In contrast, SymTrack with AIE dynamically adjusts its search region to maintain robust coverage, keeping the target centered without unnecessarily expanding the search area. This confirms that our AIE is a critical component for

robust text tracking in dynamic scenes.

**Visualization.** To intuitively demonstrate the effectiveness of our proposed method, particularly in challenging scenarios involving similar distractors, partial occlusions, and severe aspect ratio variations caused by viewpoint changes, we visualize the tracking results of SymTrack and three state-of-the-art trackers (Ye et al., 2022; Wei et al., 2023; Zheng et al., 2024) on the benchmark dataset. As shown in Figure 6, benefiting from CEC and AIE, SymTrack effectively mitigates the architectural imbalance between the backbone and prediction head, while reinforcing text-specific feature attention. This enables precise discrimination between scene text and background, achieving robust tracking performance even for small or visually complex textual targets.

## 5. Conclusion

In this work, we present the first systematic work for STT and identify three key difficulties: 1) severe distortions from perspective shifts causing feature mismatches, 2) high visual ambiguity across different instances, and 3) high sensitivity to fine-grained structural details. To address them, we propose SymTrack, a detection-free framework with a synergistic dual-branch design. It incorporates a PTR module to rectify structural imbalance, a CEC mechanism to mitigate semantic bias using textual priors, and an AIE to stabilize predictions and improve fine-grained localization sensitivity. Lacking dedicated STT benchmarks, we utilize three datasets from video text spotting domain, rigorously processing them to retain challenging, continuous trajectories. Extensive experiments demonstrate that SymTrack sets the new state-of-the-art performance, confirming the effectiveness and necessity of a specialized solution for this task. Notably, our detection-free paradigm achieves higher tracking performance than previous end-to-end pipelines that couple detection, tracking, and recognition, underscoring the advantage of a unified, tracking-centric design. We hope this work will inspire further research into detection-free STT.

## Acknowledge

This work is supported by the National Natural Science Foundation of China (Grant NO 62376266 and 62406318).

## Impact Statement

This paper presents work whose goal is to advance the field of Machine Learning. There are many potential societal consequences of our work, none which we feel must be specifically highlighted here.

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

In this appendix, we provide additional evidence and analyses to further support the claims made in the main paper. Section A contains comprehensive ablation studies and generalization experiments that validate the necessity and effectiveness of our core components. Specifically, we demonstrate the indispensable role of the dual-branch architecture, systematically analyze the hyperparameter sensitivity of the AIE, and show that the training-free AIE module can be seamlessly plugged into other state-of-the-art trackers to yield consistent gains. To address potential concerns regarding model efficiency, Section B provides a granular breakdown of the computational cost associated with each component, verifying the lightweight nature of our proposed interaction modules. Section C details the rigorous pipelines we developed to enable meaningful comparisons in the absence of prior STT benchmarks. We describe the noiseless conversion process from VTS datasets to high-quality STT benchmarks and the standardized evaluation protocol used to fairly assess end-to-end VTS models under the SOT paradigm. Finally, Section D presents extensive visualization results on challenging sequences from all benchmarks, clearly illustrating the superiority of SymTrack over recent top-performing trackers in handling fixed-position text under complex backgrounds, severe perspective distortion, dense distractors, partial occlusion, and fast motion.

## A. Design Validation and Generalization

### A.1. Ablation on Backbone Choice

To validate the necessity of the proposed dual-branch design, we evaluate a variant that relies solely on the textual feature expert. In this configuration, denoted as SymTrack (Text-Only), the main visual backbone and the PTR module are entirely removed. Both template and search region frames are processed exclusively by the textual feature expert backbone. Its output is then fed into the CEC module, which operates in a self-attention manner in this setting, and finally passed to the prediction head.

The results, reported in Table 7, clearly demonstrate the critical role of the visual backbone. Removing it leads to a severe performance collapse, with AUC on ArTVideo$_{\text{SOT}}$ dropping by **30.7%**. This substantial degradation confirms that, although the textual feature expert excels at semantic representation and is essential for calibration via CEC, it cannot substitute for the main visual backbone. The textual expert is pre-trained to extract fine-grained semantic features tailored for scene text instances, but it lacks the robust spatio-temporal reasoning and relational modeling capabilities of the visual backbone, which is pre-trained on large-scale tracking datasets.

These findings affirm that both branches are indispensable: the visual backbone provides robust spatio-temporal localization and structural awareness, while the textual feature expert supplies critical semantic calibration specific to scene text. Their synergistic integration is key to SymTrack's effectiveness on the challenging scene text tracking task.

*Table 7.* Ablation on backbone architecture. "Text-Only" removes the main visual backbone and PTR module, relying exclusively on the textual feature expert. Experiments are conducted on ArTVideo$_{\text{SOT}}$.

| Method | AUC (%) | $P_{Norm}$ (%) | $P$ (%) |
|---|---|---|---|
| Dual-Branch (Ours) | **77.74** | **91.29** | **95.88** |
| Text-Only | 47.04 | 47.33 | 66.53 |

### A.2. Ablation Study on Hyperparameters of AIE

In our AIE module, we tune three key hyperparameters: the confidence threshold $\tau_{\text{uncert}}$, the alternative scale factors $\mathcal{S}_{factors}$, and the Kalman fusion weight $\alpha_{\text{kalman}}$. To justify the choices adopted in the main paper (confidence threshold of 0.98, alternative scale factors $\{0.95, 1.05\}$, and Kalman fusion weight of 0.5), we perform systematic ablation studies by varying one parameter at a time while fixing the others to their respective optimal values. Results are reported in Table 8.

**Analysis of $\tau_{\text{uncert}}$.** As shown in Table 8, a confidence threshold of 0.98 yields the highest performance. Reducing the threshold to 0.97 triggers the multi-scale search excessively often, resulting in an AUC degradation of **0.35%**. Raising it to 0.99, conversely, prevents timely re-tracking in critical situations and incurs a larger AUC drop of **0.69%**.

**Analysis of $\mathcal{S}_{factors}$.** Table 8 shows that the scale set $\{0.95, 1.05\}$ proves optimal. Expanding the range to $\{0.90, 1.10\}$ introduces excessive noise and reduces AUC by **0.66%** relative to the best configuration. Narrowing it to $\{0.985, 1.015\}$, on the other hand, fails to adequately accommodate scale variations, leading to an AUC decrease of **0.29%**.

**Analysis of $\alpha_{\text{kalman}}$.** This weight governs the balance between the motion prior from the Kalman filter and the current-frame

response of the visual tracker. An equal weighting with Kalman fusion weight of 0.5 achieves the optimal trade-off. Favoring the motion model more heavily with Kalman fusion weight of 0.4 yields a modest AUC decline of **0.22%**, whereas overweighting the visual tracker's response at Kalman fusion weight of 0.6 impairs trajectory smoothing and causes a more pronounced drop of **0.65%**.

In summary, the ablation results confirm that a confidence threshold of 0.98, scale factors $\{0.95, 1.05\}$, and Kalman weight of 0.5 constitute the optimal hyperparameter configuration for the AIE module.

*Table 8.* Unified hyperparameter ablation table. Experiments are conducted on ArTVideo$_{SOT}$. The best setting for each hyperparameter block is highlighted in **bold**.

| Hyperparameters | | | Results | | |
|---|---|---|---|---|---|
| $\tau_{uncert}$ | $\mathcal{S}_{factors}$ | $\alpha_{kalman}$ | AUC (%) | $P_{Norm}$ (%) | $P$ (%) |
| 0.97 | $\{0.95, 1.05\}$ | 0.5 | 77.39 | 90.70 | 94.65 |
| **0.98** | $\{0.95, 1.05\}$ | 0.5 | **77.74** | **91.29** | **95.88** |
| 0.99 | $\{0.95, 1.05\}$ | 0.5 | 77.05 | 90.85 | 95.10 |
| 0.98 | $\{0.90, 1.10\}$ | 0.5 | 77.08 | 90.55 | 94.40 |
| 0.98 | **$\{0.95, 1.05\}$** | 0.5 | **77.74** | **91.29** | **95.88** |
| 0.98 | $\{0.985, 1.015\}$ | 0.5 | 77.45 | 90.93 | 95.13 |
| 0.98 | $\{0.95, 1.05\}$ | 0.4 | 77.52 | 91.05 | 95.45 |
| 0.98 | $\{0.95, 1.05\}$ | **0.5** | **77.74** | **91.29** | **95.88** |
| 0.98 | $\{0.95, 1.05\}$ | 0.6 | 77.09 | 90.90 | 94.89 |

### A.3. Generalizability of AIE

To further validate the robustness and generalizability of our training-free AIE, we apply it to two other high-performing, vision-only trackers: OSTrack (Ye et al., 2022) and ODTrack (Zheng et al., 2024). The AIE module is integrated into their inference pipeline without any re-training.

As shown in Table 9, the AIE module provides consistent and significant performance improvements for both trackers. For OSTrack$_{256}$, AIE boosts the AUC by **1.85%**, and for ODTrack, it provides a gain of **1.22%** AUC. These results demonstrate that the AIE's components, the dynamic search region adjustment and the temporal regularization model, are effective not just for SymTrack, but as a plug-and-play enhancement for other trackers. This confirms that AIE successfully stabilizes predictions and enhances robustness against the complex dynamics, such as rapid scale changes and jitter, that are common in scene text tracking.

*Table 9.* Generalizability study of the training-free AIE module on existing trackers. Experiments are conducted on ArTVideo$_{SOT}$. **Bold** indicates the improved performance with AIE, and *Italics* denote the gain.

| Method | AUC (%) | $P_{Norm}$ (%) | $P$ (%) |
|---|---|---|---|
| OSTrack$_{256}$ | 64.86 | 77.95 | 81.99 |
| OSTrack$_{256}$ + AIE | **66.71** | **80.42** | **83.57** |
| *Gain* | *+1.85* | *+2.47* | *+1.58* |
| ODTrack | 69.81 | 83.54 | 86.68 |
| ODTrack + AIE | **71.03** | **84.92** | **88.82** |
| *Gain* | *+1.22* | *+1.38* | *+2.14* |

## B. Detailed Analysis of Computational Complexity

To complement the inference analysis in the main paper, we provide a comprehensive breakdown of SymTrack's computational cost, comparing it against leading state-of-the-art trackers: SeqTrack (Chen et al., 2023), ODTrack (Zheng et al., 2024), and DUTrack (Li et al., 2025b). This analysis aims to identify computational bottlenecks, verify the efficiency of our proposed modules, and justify the trade-off between complexity and performance.

Table 10 details the architectural configuration, specialized modules, parameter count, MACs, inference speed, and tracking performance for each method.

*Table 10.* Comparative complexity analysis. We verify that SymTrack's complexity yields justified performance gains. Encoder depth denotes the depth of the main backbone. **Performance** is reported as AUC (%) on the challenging BOVText$_{SOT}$ benchmark.

| Configuration | SeqTrack$_{L\text{-}384}$ (Chen et al., 2023) | ODTrack (Zheng et al., 2024) | DUTrack$_{384}$ (Li et al., 2025b) | SymTrack (Ours) |
|---|---|---|---|---|
| Encoder depth | [MSA, 16 heads / MLP] ×24 | [MSA, 16 heads / MLP] ×24 | [MSA, 16 heads / MLP] ×24 | [MSA, 12 heads / MLP] ×12 |
| PTR Module | - | - | - | [Linear Map / 1×1 Depth-wise Conv] |
| Text Expert | - | - | - | [MSA, 64 heads / MLP] ×48 (Chen et al., 2024) |
| CEC Module | - | - | - | [Cross-Attn, 4 heads / 3×3 Conv Head] |
| Specific Design | *Seq2Seq Decoder* | *Video Token Elimination* | *Dynamic LLM* | *AIE (Training-free)* |
| Params (M) | 306.53 | 312.29 | 105.30 | 395.92 |
| FPS (Hz) | 15.88 | 32.75 | 18.57 | 22.07 |
| MACs (G) | 523.51 | 223.12 | 233.97 | 1315.00 |
| Performance (AUC) | 60.42 | 64.74 | 65.08 | **77.06** |

### B.1. Visual Encoder Efficiency.

As shown in Table 10, generic trackers like SeqTrack and ODTrack rely on scaling up the visual backbone to ViT-Large to improve feature representation. In contrast, SymTrack employs a more efficient ViT-Base as the trainable visual encoder. This lightweight foundation ensures that the core tracking mechanism remains efficient, reserving computational budget for the semantic expert.

### B.2. Impact of Text Expert.

The primary source of SymTrack's complexity is the integration of the text expert. We adopt the visual encoder of TokenFD, which is derived from InternVL (Chen et al., 2024) and further adapted on text-centric datasets. This backbone consists of a massive 48-layer encoder. While this significantly increases the computational load compared to trackers such as DUTrack, it is a necessary trade-off. As demonstrated in Section A.1, removing this expert leads to a performance collapse.

### B.3. Efficiency of Interaction Modules.

Crucially, our novel interaction modules are designed to be extremely lightweight, ensuring that the overhead is concentrated in high-value feature extraction rather than the fusion process itself.

**PTR.** Designed as a lightweight modulation network, the PTR module utilizes simple channel-wise operations and $1 \times 1$ depth-wise convolutions. It adds approximately 0.6 million parameters and has a negligible impact on computational cost, adding fewer than 2 million MACs to the pipeline.

**CEC.** Similarly, the CEC module projects features into a lower dimension of 256 before interaction. By employing a compact multi-head attention block followed by a small Convolutional Neural Network head, the parameter overhead is kept low at approximately 0.7 million, with a computational cost of around 0.12 G MACs.

### B.4. Performance vs. Complexity.

Despite the higher computational cost, SymTrack achieves a decisive performance advantage. On the BOVText$_{SOT}$ benchmark, it outperforms the computationally heavy SeqTrack-L, which requires 523 G MACs, by a margin of **16.64%** in terms of AUC. Similarly, it surpasses ODTrack and its 223 G MACs consumption by an AUC margin of **12.32%**. Furthermore, operating at an inference speed of 22.07 FPS, SymTrack remains faster than the autoregressive SeqTrack which runs at 15.88 FPS, and is comparable to the complex multi-modal DUTrack with its speed of 18.57 FPS. This confirms that our design delivers a highly efficient trade-off for high-precision scene text tracking.

## C. Benchmark Construction and Fair Evaluation

### C.1. Detailed Dataset Construction Pipeline

As stated in the main paper, we developed a systematic pipeline to convert VTS datasets into a SOT-compatible format for STT. Algorithm 1 outlines this pipeline.

Our conversion method is designed to be noiseless and rigorous, ensuring the resulting STT benchmarks fully inherit both the annotation precision and the high data quality of their VTS origins. The process is as follows:

**Trajectory Grouping.** We first parse all video annotations and group them by their unique instance ID.

**Continuity Splitting.** This is the most critical step. We sort annotations for each instance by frame ID and iterate through them. A trajectory is split into a new, separate SOT sample only when an annotation for that instance is entirely absent for one or more frames. This VTS convention typically signifies the target's complete disappearance (*e.g.*, exiting the frame). Crucially, this method retains all frames where the target is partially occluded or blurred, as long as a partial annotation is still provided. This ensures our benchmark retains challenging real-world scenarios.

**Filtering.** We discard all resulting tracklets that are shorter than a minimum threshold (5 frames), as they are too short for meaningful tracking.

**Format Conversion.** For each valid, continuous tracklet, we generate a self-contained SOT sample. We convert the source polygon annotations into minimal enclosing bounding boxes and write them to a ground truth file. To avoid massive data duplication and disk usage, we use symbolic links to create a re-indexed image sequence pointing to the original video frames.

---

**Algorithm 1** VTS to STT Dataset Curation Pipeline

---

**Input:** Video dataset $\mathcal{D}$ with annotations; minimum tracklet length $\tau_{\text{len}}$
**Output:** SOT-formatted dataset $\mathcal{D}_{\text{SOT}}$
**For** each video $v$ in $\mathcal{D}$:
   Group all annotations by instance $\{A_v^i\}$
   **For** each instance $A_v^i$:
      Sort annotations by frame ID to form tracklets $\mathcal{T}$
      **For** each tracklet $t_j$ in $\mathcal{T}$:
         **i.** Skip $t_j$ if it has fewer than $\tau_{\text{len}}$ frames
         **ii.** Convert polygon annotations of $t_j$ to bboxes
         **iii.** Re-index and link frames
         **iv.** Add $t_j$ to $\mathcal{D}_{\text{SOT}}$
**Return** $\mathcal{D}_{\text{SOT}}$

---

### C.2. VTS Evaluation Pipeline

To ensure a fair comparison with VTS models (Wu et al., 2025; 2024a) in the main paper, a standardized conversion pipeline is required. VTS models report their results in a consolidated format that bundles all tracked instances of an entire video into a single container, whereas standard SOT evaluation expects one individual prediction file per tracklet. We therefore implement a rigorous conversion pipeline, summarized in Algorithm 2, to bridge this format discrepancy.

The pipeline proceeds as follows. First, we parse the consolidated VTS output to extract all object trajectories and organize them into a data structure that associates each unique object ID with its corresponding frame-level bounding boxes. A critical difference lies in the fact that VTS models only provide bounding boxes for frames in which the object is actively recognized. In contrast, SOT evaluation protocols require a prediction in every frame of the sequence. To satisfy this requirement, missing annotations are filled by propagating the most recent valid bounding box forward. For objects that do not appear (or are not recognized) in the initial frame, a zero-area placeholder box is inserted until the first valid annotation. Each object's complete, gap-filled trajectory is then written to a separate text file named according to its object ID. Finally, these generated prediction files are automatically aligned with the official SOT ground truth tracklets for standardized evaluation.

This systematic conversion process guarantees that VTS approaches, despite their fundamentally different output conventions, are evaluated under exactly the same conditions as other methods, enabling direct and fair performance comparison.

### C.3. ID-Agnostic Evaluation of VTS Baselines

To further examine whether the large gap between SymTrack and VTS-based methods merely results from trajectory discontinuity or identity-association penalties, we additionally evaluate VTS baselines under a maximally forgiving ID-agnostic protocol. Specifically, for each frame, we disregard the track ID predicted by the VTS model and match the

---

**Algorithm 2** Conversion Pipeline from VTS Output to Standard SOT Evaluation Format

---

**Input:** Directory containing VTS results, VTSResultsDir

**Output:** Directory for SOT-formatted predictions, SOTPredictionDir

**For** each video in VTSResultsDir:

    Parse the consolidated VTS result to extract all object trajectories and the total number of frames, FrameCount

    Create an output subdirectory for the video, VideoPredictionDirectory

    **For** each object instance with id ObjectID and detection data in the trajectories:

        **i.** Initialize a full-frame trajectory list, FullTrajectory, of size FrameCount

        **ii.** Fill missing frames in FullTrajectory by propagating the last known bounding box forward (use a zero-area box before the first valid annotation)

        **iii.** Save the completed FullTrajectory as a separate text file in VideoPredictionDirectory

**Return** SOTPredictionDir

---

ground-truth target with the predicted bounding box that yields the highest IoU. This setting neutralizes penalties caused by ID switches, missing associations, or broken trajectories, and therefore approximates an upper-bound localization assessment for VTS systems under the STT evaluation setting.

*Table 11.* ID-agnostic evaluation of VTS baselines. "Strict ID" denotes the standard conversion-based SOT evaluation used in the main paper, while "ID-Agnostic" selects the highest-IoU prediction in each frame regardless of the predicted identity. $OP_{75}$ denotes overlap precision at IoU threshold 0.75.

| Method | Protocol | ArTVideo$_{SOT}$ | | DSText$_{SOT}$ | |
|---|---|---|---|---|---|
| | | AUC (%) | $OP_{75}$ (%) | AUC (%) | $OP_{75}$ (%) |
| TransDETR | Strict ID | 9.18 | – | 5.08 | – |
| TransDETR | ID-Agnostic | 41.10 | 17.59 | 33.32 | 16.13 |
| TransVTSpotter | Strict ID | 8.84 | – | 4.93 | – |
| TransVTSpotter | ID-Agnostic | 39.89 | 16.76 | 31.19 | 15.47 |
| **SymTrack (Ours)** | Standard | **77.74** | **76.57** | **70.66** | **58.68** |

Even under this identity-neutralized protocol, the VTS baselines remain substantially behind SymTrack. This result indicates that the observed gap is not solely an artifact of broken trajectories or identity conversion, but also reflects the limited frame-wise localization precision of detection-driven VTS systems under challenging text dynamics. Accordingly, the VTS comparison could be regarded as a task-paradigm mismatch, while the ID-agnostic results further confirm the advantage of continuous, target-specific tracking for STT.

### C.4. Practical Initialization in General Videos

STT is formulated as an instance-oriented tracking task that requires the target text region in the first frame. In practical applications, this initialization can be obtained either through a human-in-the-loop interface, such as click-to-edit interaction, or through an automated first-frame localization module. For general videos, a practical deployment pipeline is *Text Grounding + Global Tracking*: the user provides a natural-language description of the target text instance, a grounding tool (Rong et al., 2017; Li et al., 2025a) processes only the first frame to produce the initial text box, and SymTrack then maintains the target trajectory across all subsequent frames. This avoids expensive and fragile per-frame detection while preserving continuous target-specific tracking. Furthermore, even if the grounding model's initial box includes background deviations, our CEC structurally locks onto the intrinsic text strokes, and the PTR actively filters out the trapped noise. This pipeline perfectly bridges automated initialization with high-precision tracking in videos.

### C.5. Discussion and Additional Evaluation on Earlier Video Text Benchmarks

We also discuss earlier video-text resources highlighted in prior literature. ICDAR 2013 Robust Reading Competition (Karatzas et al., 2013) includes a video text component, but the test annotations required for a complete STT-style conversion are not publicly available in a form that supports rigorous benchmark construction.

To nevertheless examine robustness on an earlier resource with usable annotations, we conduct zero-shot evaluation on the ICDAR 2013 video training set. As shown in Table 12, SymTrack maintains a clear advantage over both ODTrack and

DUTrack, indicating that its gains are not restricted to the three curated modern benchmarks.

*Table 12.* Zero-shot evaluation on the ICDAR 2013 video training set.

| Type | Method | AUC (%) | $P_{\text{Norm}}$ (%) | $P$ (%) |
|---|---|---|---|---|
| Vision-only | **SymTrack (Ours)** | **63.87** | **82.27** | **76.83** |
| Vision-only | ODTrack | 56.11 | 72.78 | 67.89 |
| Vision-language | DUTrack | 54.59 | 70.80 | 66.24 |

## D. Qualitative Analysis

In order to visually highlight the advantages of our method over existing approaches in challenging scenarios, we provide additional visualization results in Figure 7 and Figure 8. We compare our proposed SymTrack with OSTrack (Ye et al., 2022), ODTrack (Zheng et al., 2024) and ARTrack (Wei et al., 2023) in terms of performance when the target undergoes large perspective-induced deformation and scale variation, partial occlusion, dense text with high instance similarity, and fast-moving or partially occluded text that demands precise localization of fine-grained structures. All videos are taken from the challenging DSText$_{\text{SOT}}$ and BOVText$_{\text{SOT}}$ benchmarks.

**Fixed-position text under complex backgrounds.** In Figure 7, subtitles and game overlays typically remain static relative to the screen but undergo extreme background variations, camera shake, or motion blur. As identified in the main paper, generic trackers lack text-specific priors and treat these instances as ordinary rigid objects, making them vulnerable to strong background distractors. Once the correlation peak is contaminated, they irreversibly drift to irrelevant regions (*e.g.*, OSTrack locks onto moving players, ARTrack jumps to UI elements with similar colors). Benefiting from the textual feature expert and CEC, SymTrack injects strong semantic priors that explicitly calibrate the feature map toward textual regions, enabling the model to ignore drastic background changes and maintain focus on the target characters throughout the entire sequence.

**Extreme distortion, dense text, and fast motion.** Figure 8 highlights the three core difficulties of scene text tracking simultaneously: 1) *Severe perspective shifts* cause dramatic geometric distortions and scale variation. The PTR module proactively refines the high-entropy visual features before they reach the prediction head, eliminating the structural bottleneck that plagues vanilla transformer-based trackers. 2) *High visual ambiguity* in dense text scenes creates powerful distractors with nearly identical local patterns. The CEC branch provides precise textual priors, allowing SymTrack to unambiguously distinguish the target instance from dozens of similar candidates. 3) *Fine-grained structural sensitivity* combined with fast motion or occlusion demands robust temporal modeling. The AIE dynamically adjusts search scales and applies Kalman-based smoothing only when necessary, preventing jitter and long-term drift that affect all competing methods.

These qualitative results strongly corroborate our quantitative gains and validate that the proposed PTR, CEC, and AIE modules synergistically address the unique challenges of scene text tracking, enabling SymTrack to achieve unprecedented robustness where generic trackers fail.

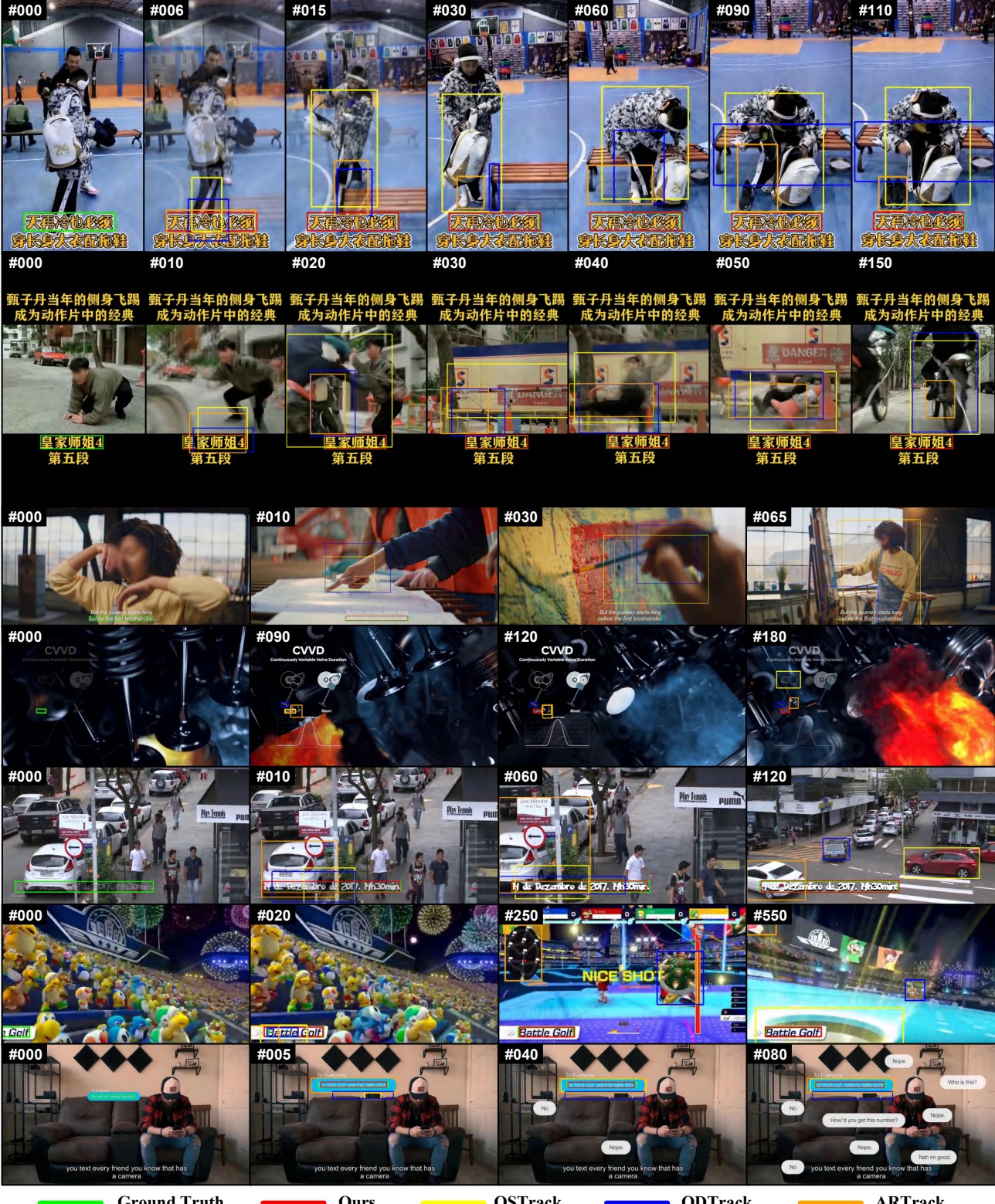

*Figure 7.* Visual comparison in scenes with relatively fixed-position text (*e.g.*, subtitles, game UI, and overlaid captions) under drastic background changes or heavy motion blur. Generic trackers quickly drift to background objects or similar-looking regions due to the absence of text-specific semantic modeling. In contrast, SymTrack consistently focuses on the target text throughout the sequence, owing to the CEC mechanism that effectively suppresses visual ambiguity and reinforces text-aware attention. Zoom in for details.

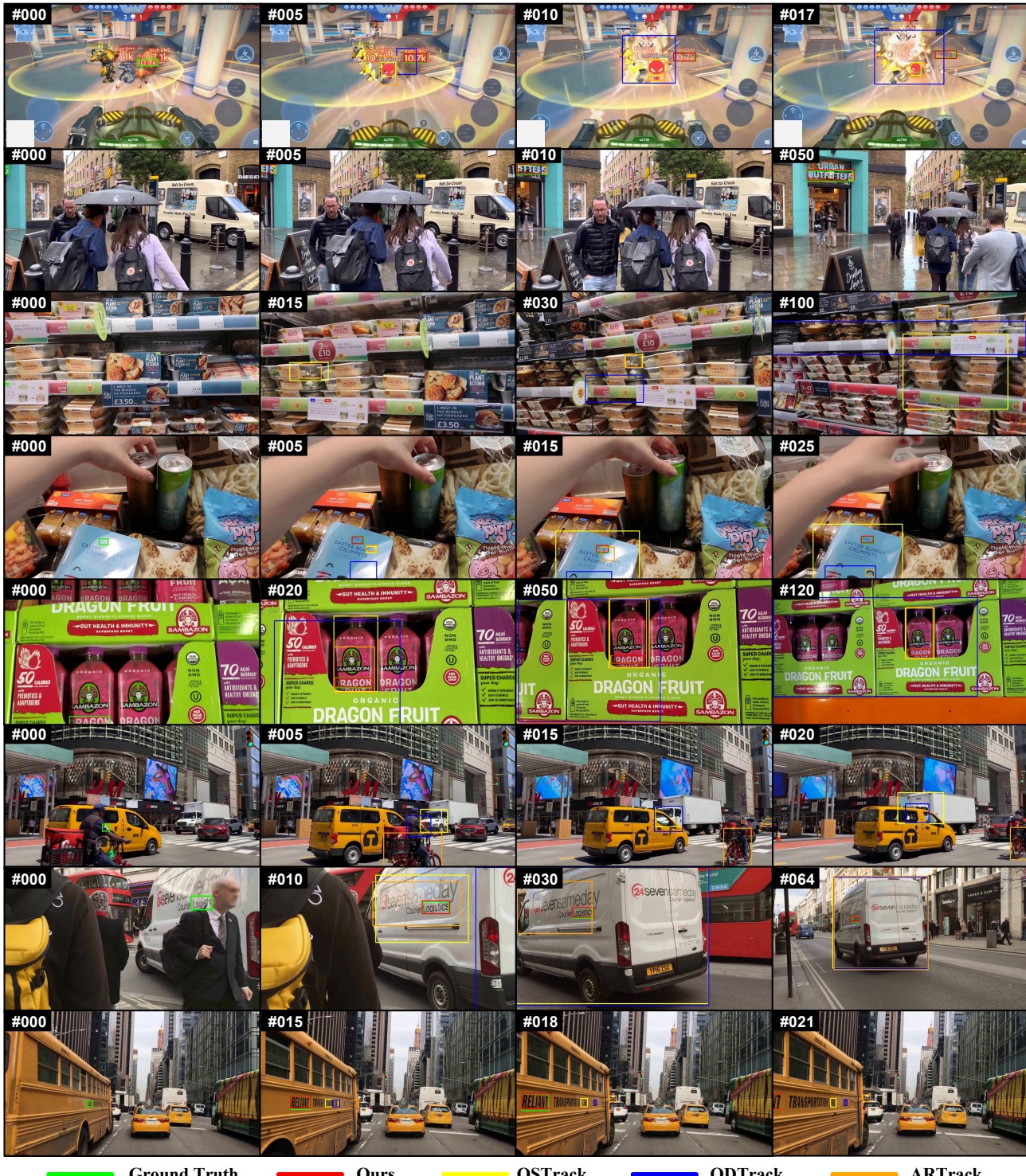

*Figure 8.* Visual comparison in highly challenging scenarios involving (a) dense and tiny scene text, (b) severe perspective-induced distortion, partial occlusion, and (c) text attached to fast-moving objects (*e.g.*, on clothing or vehicles). Competing methods either lose the target rapidly or drift to nearby distractors because of structural imbalance and insufficient semantic discrimination. SymTrack maintains accurate and stable localization in all cases, benefiting from the combined strengths of PTR that alleviates severe distortions caused by perspective shifts, CEC that resolves high visual ambiguity among similar instances, and the AIE that enforces temporal smoothness under large motion and deformation. Zoom in for details.

