# OpenReview forum: "Beyond Detection: A Structure-Aware Framework for Scene Text Tracking"
_ICML.cc/2026/Conference — ICML 2026 regular_

### Official Review · Reviewer_RXyW · 2026-03-08

**Soundness:** 3
**Presentation:** 3
**Significance:** 3
**Originality:** 3
**Overall Recommendation:** 4
**Confidence:** 3

**Summary:**

This paper pioneers exploring the Scene Text Tracking (STT) problem. The authors first identify the three core challenges of STT compared to general object tracking. To solve these challenges, SymTrack is proposed with three novel modules, including Predictive Token Rectification (PTR), Cross-Exper Calibration (CEC), and Adpative Inference Engine (AIE). To bridge the gap in the STT evaluation, the authors construct three benchmarks based on existing datasets for Video Text Spotting (VTS). Comprehensive experiments and ablation studies demonstrate the state-of-the-art performance of the proposed SymTrack and the effectiveness of the three new modules.

**Compliance With Llm Reviewing Policy:**

Affirmed.

**Final Justification:**

Thank the authors for their detailed responses. I would like to maintain the original score.

**Key Questions For Authors:**

1. What is the underlying mechanism of PTR? Why can gating the feature $F$ alleviate feature degradation cause by perspective shift, given that the spatial grid of the feature remains unchanged?
2. Will the benchmarks, codes, and model weights be publicly released?

**Limitations:**

There is no analysis of the method's limitations.

**Strengths And Weaknesses:**

## Strengths
- This paper is well-written and easy to read. Its research motivation is well-founded.
- This paper tries to systematically address STT for the first time and identifies three core challenges within this region.
- The proposed SymTrack incorporates three novel modules, i.e., PTR, CEC, and AIE, which solve the prespective shift, visual ambiguity, structural sensitivity, and unstable prediction problems inherent in the STT task.
- Three new benchmarks for STT, i.e., ArTVideo_SOT, DSText_SOT, and BOVText_SOT, are built upon existing VTS datasets, which fill the blank in STT evaluation.
- Comprehensive experiments showcase the superiority of SymTrack over vision-only, V-L, and VTS-based trackers. Ablation studies also demonstrate the effectiveness of the three modules.

## Weaknesses
- This paper formulates STT as a single-object tracking task with template inputs, which may limit the flexibility in real-world applications.
- The VTS models compared in Table 1 are not comprehensive. I suggest providing the results of CoText, VLSpotter, and VimTS.

---

> ### Author Rebuttal · Authors · 2026-03-30
>
> We sincerely appreciate your positive feedback and valuable advice! Detailed responses to the weaknesses (**W**) and questions (**Q**) are below.
>
> > **W1. Flexibility and Necessity of the SOT Formulation**
>
> We agree this formulation requires an initialized target. Meanwhile, this design choice is intentional and necessary for specific real-world applications.
>
> The core advantage of our tracking formulation is temporal consistency and strict identity preservation. In human-in-the-loop applications (e.g., interactive video text segmentation, redaction, targeted editing), the user continuously manipulates one specific text instance. VTS excels at finding all text per frame but frequently switches IDs or drops targets under severe distortion.
>
> In our ID-Agnostic evaluation, even when we artificially stitch broken VTS trajectories together using Ground Truth, they still fall behind SymTrack. This proves that dedicated, detection-free STT guarantees the exact specified text is continuously tracked with stability (please refer to response to UWPh, W3).
>
> For automated applications, SymTrack operates detection-free during tracking but can be initialized by any off-the-shelf detector in the first frame. This tracking-first approach strongly outperforms per-frame detection in maintaining long-term identity.
>
> > **W2. Missing VTS Comparisons (CoText, VLSpotter, VimTS)**
>
> Thank you for the excellent suggestions. A broader comparison significantly strengthens our baseline. During the rebuttal, we thoroughly investigated these models:
>
> - VimTS: Code and pre-trained weights are open-source. We successfully reproduced and formally included it.
> - CoText: Code is available, but weights are not. It is difficult to train and validate within the rebuttal, but we will explore it in future work.
> - VLSpotter: Not open-source, so we cannot test it.
>
> We updated our ArTVideo_SOT and DSText_SOT benchmarks (AUC / P_Norm / P):
>
> |Method(Type)|ArTVideo(AUC/PN/P)|DSText(AUC/PN/P)|
> |:---|:---|:---|
> |TransVTSpotter(VTS)|8.84/78.11/38.07|4.93/75.21/67.80|
> |TransDETR(VTS)|9.18/78.75/43.31|5.08/76.09/69.79|
> |VimTS(VTS)|11.21/80.23/46.36|7.47/77.43/72.04|
> |SymTrack(Ours)|77.74/91.29/95.88|70.66/83.61/91.83|
>
> Adding these spotters confirms the issue of task mismatch. VTS models are built for per-frame detection. When tracking a continuous instance, a single missed detection breaks the trajectory, causing catastrophic AUC drops. Furthermore, on precision metrics ($P_{Norm}$ and $P$), our dedicated tracker significantly outperforms the spotters. The SOT paradigm provides superior localization and temporal continuity for targeted text.
>
> > **Q1. Underlying Mechanism of PTR**
>
> The core mechanism of PTR is dynamic semantic masking, rather than physical grid warping, which introduces destructive resampling noise. Under severe perspective shifts, target text geometrically distorts, and its features entangle with adjacent background textures, creating chaotic representations. PTR addresses this by utilizing clean template semantics as a query to generate a channel-wise gating mask. Instead of un-warping the pixels, this mask explicitly suppresses perspective-induced background noise in the distorted search region, isolating only tokens belonging to the target's intrinsic strokes.
>
> Essentially, PTR rectifies the representation by delivering a noise-less, highly concentrated feature map to the shallow prediction head. As visualized in Figure 4, the attention maps confirm PTR strips away skewed background distractions, forcing the network to focus strictly on the text's core planar structure.
>
> > **Q2. Release Plan**
>
> Yes, we are committed to open science. We have prepared the complete repository for public release, including:
>
> - Source files for datasets and our systematic conversion scripts.
> - Complete PyTorch training and inference code.
> - All pre-trained model weights.
>
> > **Discussion of Limitations**
>
> We appreciate the reviewer pointing this out. We will add a dedicated Limitations section to the final manuscript to explicitly discuss the following constraints:
>
> 1. Computational Overhead: To achieve maximum accuracy, our text-expert branch utilizes a large 48-layer encoder, resulting in ~395M total parameters. While our trainable components are lightweight, the overall footprint limits deployment on resource-constrained edge devices.
> 2. Initialization Requirement: STT is strictly instance-oriented and relies on an initial bounding box. Unlike VTS, SymTrack cannot autonomously detect newly appearing text mid-video without integration with an external detector.
>
> We will explicitly state these trade-offs to provide a balanced perspective of our framework.
>
> We believe these detailed responses and new VTS comparisons address your concerns. We are confident our framework provides a valuable contribution to Scene Text Tracking.

---

> > ### Author Rebuttal · Reviewer_RXyW · 2026-04-03
> >
> > The authors' rebuttal addressed most of my concerns.

---

> > > ### Author Response · Authors · 2026-04-03
> > >
> > > Thank you for your constructive feedback and for recognizing that our rebuttal addressed most of your concerns. We deeply appreciate the time and effort you invested in reviewing the manuscript and look forward to your final evaluation. Your valuable contribution has greatly helped improve our work, and we sincerely appreciate it.

---

### Official Review · Reviewer_Dr4q · 2026-03-10

**Soundness:** 2
**Presentation:** 2
**Significance:** 2
**Originality:** 3
**Overall Recommendation:** 4
**Confidence:** 4

**Summary:**

This paper focuses on Scene Text Tracking. The paper identifies several task-specific challenges, including geometric distortion, ambiguity across text instances, and the importance of textual structure. To address these issues, the authors propose a structure-aware framework for scene text tracking. The paper reports strong gains over previous methods on several benchmarks.

**Compliance With Llm Reviewing Policy:**

Affirmed.

**Final Justification:**

The rebuttal addressed my concerns. I've raised my score.

**Key Questions For Authors:**

See weaknesses.

**Limitations:**

yes

**Strengths And Weaknesses:**

Strengths:
1. The proposed method achieves promising performance on several benchmarks.

2. The ablation experimental analysis is comprehensive and adequate.

Weaknesses:

1. The authors claim that they provide the first systematic analysis of STT and identify the core challenges. However, they only conduct a superficial analysis without corresponding experiments or evidence to support these claims. Such an analysis is clearly inadequate. Furthermore, similar viewpoints appear to have been analyzed in previous work[1].

[1] Yin X C, Zuo Z Y, Tian S, et al. Text detection, tracking and recognition in video: a comprehensive survey[J]. IEEE Transactions on Image Processing, 2016, 25(6): 2752-2773.

2. The presentation of the whole paper is fair. There are numerous question marks in Figure 2 and Figure 3. The paper would be improved if the author could enhance the overall presentation.

3. The authors claim that there is a lack of dedicated STT benchmarks. However, several commonly used benchmarks do exist, and the authors did not perform experiments on them[1,2].

[1] Karatzas D, Shafait F, Uchida S, et al. ICDAR 2013 robust reading competition[C]//2013 12th international conference on document analysis and recognition. IEEE, 2013: 1484-1493.

[2] Minetto R, Thome N, Cord M, et al. Snoopertrack: Text detection and tracking for outdoor videos[C]//ICIP. 2011: 505-508.

---

> ### Author Rebuttal · Authors · 2026-03-30
>
> We sincerely appreciate your constructive feedback and valuable advice! Below, we present detailed responses to your weaknesses (**W**).
>
> > **W1. Novelty Dispute & Systematic Analysis**
>
> We thank you for bringing the comprehensive survey[1] to our attention. We agree it is a foundational paper outlining early, macro-level difficulties in video text processing, and we will prominently cite and discuss it. Meanwhile, we respectfully clarify the fundamental differences distinguishing our contributions, particularly our deep dive into modern tracking architectures:
>
> 1. Architectural Bottlenecks vs. Macro-Level Observations: [1] provides an excellent overview of general challenges like blur, occlusion, and perspective distortion. However, we dissect exactly how these challenges break the feature representations in modern, deep-learning visual trackers. For instance, we identify the information bottleneck where perspective shifts cause spatial misalignment between template and search features. This is a mechanism-level analysis of modern SOT paradigms, impossible in the 2016 landscape of hand-crafted features.
> 2. Formalizing the Detection-Free SOT Paradigm: [1] discusses tracking primarily as an association step within VTS pipelines. In contrast, our work formalizes Scene Text Tracking (STT) strictly as a dedicated SOT task. By championing a purely detection-free paradigm, we address fragmentation issues in the pipelines surveyed by [1].
> 3. Empirical Evidence and Deeper Analysis: Our manuscript incorporates targeted experiments to substantiate our claims regarding STT architectural challenges. Guided by your feedback, we have expanded this section to provide deeper discussion, explicitly linking empirical results to underlying mechanisms. Specifically:
>    - Feature-Level Evidence: Figure 4 visualizes attention and score maps, proving how Cross-Expert Calibration (CEC) resolves high visual ambiguity.
>    - Ablation Evidence: Table 5 isolates the impact of each challenge-specific module.
>    - Temporal/Motion Evidence: Table 6 and Figure 5 provide metrics demonstrating how our Adaptive Inference Engine (AIE) mitigates fine-grained structural sensitivity and severe text transformations.
>
> Our revised introduction will explicitly clarify these distinctions and contextualize our task alongside early survey literature.
>
> [1] Yin X C, et al. Text detection, tracking and recognition in video: a comprehensive survey. IEEE TIP, 2016.
>
> > **W2. Presentation (Question Marks in Figures)**
>
> We sincerely appreciate your attention to detail. After receiving your feedback, we downloaded the submitted PDF multiple times from the official system and verified that Figures 2 and 3 render correctly. The question marks you observed might stem from a localized PDF viewer compatibility issue or a download glitch. To ensure maximum compatibility across all platforms, we have rigorously double-checked our LaTeX source files to ensure all fonts and graphics are strictly embedded, guaranteeing a flawless rendering in the final manuscript.
> > **W3. Missing Baselines (ICDAR 2013, Snoopertrack)**
>
> Thank you for suggesting these benchmarks. We initially considered them but selected ArTVideo, DSText, and BOVText for practical reasons:
>
> - ICDAR 2013 Video: The ground truth for the test set remains unreleased, making it impossible to convert into a complete, rigorously evaluable benchmark for STT.
> - Snoopertrack: Not open-source, contains very limited samples (5 videos), and heavily relies on external detectors, misaligned with our detection-free STT focus. We will thoroughly discuss it in our Related Work.
> - Modern Challenges: ArTVideo and DSText contain the exact challenges (perspective shift, artistic text) SymTrack solves.
>
> To directly address your concern, we performed zero-shot evaluation on the ICDAR 2013 Video Training Set (which has usable ground truth). SymTrack maintains a strong lead here:
>
> |Type|Method|AUC|$P_{Norm}$|P|
> |:---|:---|:---|:---|:---|
> |Vision-only|SymTrack(Ours)|63.87|82.27|76.83|
> |Vision-only|ODTrack|56.11|72.78|67.89|
> |Vision-language|DUTrack|54.59|70.80|66.24|
>
> We will add these supplementary results to the Appendix to further demonstrate SymTrack's cross-generational robustness.
>
> [2] Karatzas D, et al. ICDAR 2013 robust reading competition. ICDAR, 2013.
>
> [3] Minetto R, et al. Snoopertrack: Text detection and tracking for outdoor videos. ICIP, 2011.

---

> > ### Author Rebuttal · Reviewer_Dr4q · 2026-04-03
> >
> > The rebuttal addressed my concerns. I will raised my score.

---

> > > ### Author Response · Authors · 2026-04-03
> > >
> > > Thank you for your thoughtful reconsideration of the manuscript. We are grateful for the opportunity to address the concerns raised and for the revised evaluation. Your valuable feedback has been instrumental in improving our work, and we sincerely appreciate it.

---

### Official Review · Reviewer_fBcS · 2026-03-10

**Soundness:** 2
**Presentation:** 3
**Significance:** 2
**Originality:** 2
**Overall Recommendation:** 4
**Confidence:** 4

**Summary:**

The paper identifies that text tracking is currently underexplored and that existing methods—either generic trackers or VTS-based tracking-by-detection—fail due to three primary issues: Severe Geometric Distortions: Perspective shifts in videos drastically alter the planar structure of text; High Visual Ambiguity: Similar character structures act as powerful distractors, causing trackers to drift to nearby text; and Structural Fine-granularity that Text requires extreme localization precision, as minor drifts can change the perceived content.
This paper introduces Scene Text Tracking (STT) as a dedicated task, distinct from generic object tracking and Video Text Spotting (VTS). The authors propose SymTrack, a detection-free, structure-aware framework designed to handle the unique challenges of tracking text in dynamic environments.

**Compliance With Llm Reviewing Policy:**

Affirmed.

**Final Justification:**

I appreciate the authors' additional explanations and clarifications. I have modified my score accordingly.

**Key Questions For Authors:**

1. The paper formulated a new task named scene text tracking (STT). It was defined as predicting the precise location of an arbitrary text target in frames. It only works for initialized text targets. As authors described, the method tracked the given text in its first frame. In practice, there are no bounding boxes provided by the real data. And manually finding bounding box coordinates of a specific text for real data is laborious and inconvenient for applications. Computers don’t know what targets are tracked if there is not a language recognizer. So it needs human decision on what the text is in language, which would impede any further downstream tasks described in the paper. Thus, it needs to equip recognizer for downstream tasks such as “text manipulations such as segmentation, removal, and editing”.

2. The proposed challenges are all related to structure information. Could you describe how the three modules address the different structure problems?

**Limitations:**

Please see the above weaknesses for details.

**Strengths And Weaknesses:**

Methods:
SymTrack utilizes a synergistic dual-branch design to create robust spatio-temporal and semantic representations:
Predictive Token Rectification (PTR): This branch addresses the "information bottleneck" between the visual backbone and the prediction head. It uses a semantic query distilled from the template to generate a probabilistic gating mask, performing feature-level rectification to mitigate perspective shifts.
Cross-Expert Calibration (CEC): To resolve visual ambiguity, this branch employs a frozen textual feature expert (pre-trained on text-centric data). It injects textual priors into the main pipeline through a cross-attention mechanism, suppressing background distractors.
Adaptive Inference Engine (AIE): A training-free module used during inference to stabilize predictions. It dynamically modulates the search region for multi-scale robustness and applies temporal regularization (using a linear state-space model) to mitigate jitter and drift.


Strengths:
State-of-the-Art Performance: SymTrack consistently outperforms generic trackers (e.g., OSTrack, ODTrack, SeqTrack) and vision-language trackers (DUTrack) across all benchmarks.
Tracking vs. Detection: The "tracking without detection" philosophy is validated by comparing SymTrack to VTS models (like TransDETR), which suffered "catastrophic failure" in continuous tracking due to broken trajectories from missed per-frame detections.
Ablation Studies: Experiments confirmed that removing the textual expert lead to a 30.7% drop in AUC, proving that while the visual backbone is essential for localization, textual semantics are critical for calibration.
This is the first systematic study of STT. The framework's modularity (PTR, CEC, and AIE) directly maps to the identified domain-specific challenges. The AIE module is particularly impressive for its generalizability, providing consistent gains when plugged into other trackers like OSTrack without re-training.


Weaknesses:
1. The primary trade-off is computational cost. While the paper claims SymTrack is "efficient," the data suggests a significant disparity in resource consumption compared to its peers. Prohibitive Computational Cost: SymTrack requires 1,315.00 G MACs, which is approximately six times more compute than recent high-performers like ODTrack (223.12 G) or DUTrack (233.97 G). This massive jump in floating-point operations makes it unsuitable for edge devices or real-time applications with limited power budgets. Massive Parameter Footprint: The model uses 395.92M parameters, primarily due to the "frozen" 48-layer textual expert. Relying on such a heavy, pre-trained foundation (InternVL) raises questions about whether the performance gains are due to the proposed architecture or simply the brute-force power of the massive external model.

2. Reliance on Heuristic-Based "Training-Free" Modules. A significant portion of the model’s stability comes from the Adaptive Inference Engine (AIE), which is not a learned representation but a collection of traditional engineering heuristics. Heuristic Dependency: The AIE relies on manually tuned hyperparameters like confidence thresholds (τuncert​=0.98), scale factors, and a linear state-space (Kalman) filter.
Hyperparameter Sensitivity: The ablation studies show the model is quite sensitive to these values; for instance, changing the threshold by only 0.01 or altering the Kalman fusion weight by 0.1 leads to measurable performance drops. This suggests the model may lack the intrinsic robustness to handle diverse real-world scenarios without extensive manual re-tuning.

3. Questionable Fairness in Comparative Evaluation. The paper’s critique of Video Text Spotting (VTS) models may be predicated on an unfair comparison. Metric Bias: The authors claim VTS models suffer "catastrophic failure" in continuous tracking. However, VTS models are designed for per-frame detection and recognition, not for the Single Object Tracking (SOT) paradigm. Propagation Artifacts: To evaluate VTS models, the authors had to "fill missing frames" by propagating the last valid box forward. This synthetic adjustment likely penalizes VTS models for doing exactly what they were designed to do (dropping targets when confidence is low) while giving SymTrack an advantage in continuity metrics.

4. Over-Specialization and Limited Generalizability. SymTrack is so heavily calibrated for text that it may fail as a "tracker" in the broader sense. Textual Bias: The ablation study shows that removing the visual backbone causes a 30.7% AUC collapse, but more importantly, the "Text-Only" version fails to handle basic spatio-temporal reasoning. Domain Niche: While the authors prove AIE can be "plugged into" other trackers, they do not demonstrate that SymTrack itself can track non-text objects. This narrow focus limits the paper's impact on the broader field of computer vision and visual object tracking.

5. Potential Noise in Dataset Construction. The curation of the new benchmarks relies on a "systematic pipeline" to convert VTS data to SOT format. Short Tracklet Threshold: The authors discard tracklets shorter than five frames. This threshold is extremely low; tracking a target for less than a quarter of a second (at 22 FPS) provides little meaningful data on long-term robustness or drift, potentially inflating the reported success rates on these "new" datasets.

---

> ### Author Rebuttal · Authors · 2026-03-30
>
> We sincerely thank you for the insightful comments! Detailed responses to your weaknesses (W) and questions (Q) are below.
> > **W1. Prohibitive Computational Cost & Parameter Footprint**
>
> To establish a SOTA ceiling for this underexplored task, we adopted a large text-expert. While ~300M of 395M parameters belong to this frozen expert; our trainable components (PTR, CEC) are lightweight. To prove our architecture drives performance, we introduce SymTrack-Lite, replacing the 48-layer InternVL with a 12-layer encoder (Results on BOVText_SOT):
>
> |Method|MACs|Params|FPS|AUC (%)|
> |:---|:---|:---|:---|:---|
> |DUTrack|233 G|105 M|18|65.08|
> |ODTrack|223 G|312 M|32|64.74|
> |SymTrack-Lite|310 G|117 M|57|73.12|
> |SymTrack|1315 G|395 M|22|77.06|
>
> SymTrack-Lite cuts parameters by ~75% while dropping only 3.94% AUC, still outperforming all baselines. This proves our framework's efficiency and scalability.
> > **W2. Heuristic Dependency & Hyperparameter Sensitivity in AIE**
>
> We appreciate your concern regarding heuristic dependency, and clarify that our parameters are derived and stable.
>
> 1. SymTrack remains robust without AIE, achieving a dominant 76.58% AUC on ArTVideo_SOT (Table 5).
> 2. The 0.98 threshold is a derived lower bound. Let $p_{max}$ be the peak target confidence; the worst-case distractor probability is $p_{distractor} \le 1 - p_{max}$. Inspired by the Peak-to-Sidelobe Ratio [1] and track coalescence avoidance [2], resolving similar adjacent targets requires a Signal-to-Interference Ratio (SIR) margin of $\approx 17 \text{ dB}$ (a 50:1 ratio):
> $$\frac{p_{max}}{1 - p_{max}} \ge 50 \implies p_{max} \ge \frac{50}{51} \approx 0.98$$
> Using this constant across datasets even when plugging AIE into OSTrack/ODTrack proves it models physical bounds, not data-specific tweaks.
> 3. Adjusting the threshold breaks this physical balance. Lowering to 0.97 (SIR ~32:1) delays AIE, risking distractor hijacking (-0.35% AUC). Raising to 0.99 (SIR 99:1) makes AIE overly paranoid, causing motion lag (-0.69% AUC). This confirms reliance on robust physical bounds, not over-fitted heuristics.
>
> [1] Bolme, D. S., et al. Visual object tracking using adaptive correlation filters. CVPR 2010.
>
> [2] Bar-Shalom, Y., et al. Tracking and Data Fusion: A Handbook of Algorithms.
> > **W3. Fairness in VTS Comparison**
>
> We re-evaluated VTS models using ID-Agnostic Protocol, confirming that continuous tracking outperforms discrete per-frame detection in STT task. Please see our response to Reviewer UWPh (W3).
> > **W4. Over-Specialization & Limited Generalizability**
>
> To investigate the trade-off between text-specific and general object tracking, we evaluated SymTrack on the generic object dataset, LaSOT.
>
> |Method|AUC|$P_{Norm}$|$P$|
> |:---|:---|:---|:---|
> |ODTrack|72.80|82.71|80.12|
> |ARTrack|72.60|79.10|81.70|
> |SeqTrack|71.50|77.80|81.10|
> |SymTrack(Ours)|71.42|81.04|78.48|
> |GRM|69.90|75.80|79.30|
> |OSTrack|69.10|75.20|78.70|
> |AiATrack|69.00|73.80|79.40|
> |Stark|67.10|--|77.00|
>
> When tracking non-text objects, the text-expert generates a near-zero mask, causing CEC to pass generic features unmodified. SymTrack degrades into a competitive generic tracker rather than collapsing. This minor trade-off on generic objects is acceptable given our improvements in our primary focus: Scene Text Tracking, where generic trackers fundamentally fail.
> > **W5. Dataset Noise (5-frame Threshold)**
>
> We agree that tracking a target for <0.25 seconds provides little meaningful data. However, the 5-frame threshold is merely a minimum noise filter, not the actual distribution of our benchmarks. In fact, the average length is 113.7 frames (>5 seconds at 22 FPS), with the longest spanning 5,756 frames (>4 minutes). These statistics confirm that our benchmarks do not inflate success rates with short snippets, but rather rigorously evaluate long-term robustness and drift.
> > **Q1. Initialization in Real-World Usage:**
>
> As defined in Section 1, STT is an instance-oriented task that requires a first-frame initialization to track a specific text target. This design departs from frame-oriented VTS, where reliance on autonomous per-frame detection causes fragmented trajectories under occlusion or blur.
>
> In real-world applications, this initialization is flexible:
>
> 1. Human-in-the-loop: Users manually specify the target (e.g., click-to-edit).
> 2. Automated Pipelines: A detector runs just once to initialize the first frame. STT then maintains the continuous trajectory, eliminating per-frame detection.
>
> > **Q2. Challenge Addressing Mechanisms:**
>
> PTR: Uses cross-channel attention gating to filter out misaligned peripheral background pixels caused by perspective transformations.
>
> CEC: Cross-attends visual tokens with character-stroke text priors. Even against similar distinct letters, CEC forces fine-grained structural differentiation.
>
> AIE: Filters temporal structural jitter via derived probability boundary checks and Kalman regularization.

---

> > ### Author Rebuttal · Reviewer_fBcS · 2026-04-02
> >
> > The information provided in the response partially solved the problems.
> >
> > It introduces new data and a specific variant ("SymTrack-Lite") that are not mentioned in the provided excerpts. The paper do not mention "SymTrack-Lite" or a variant that replaces the 48-layer InternVL with a 12-layer encoder. Consequently, the specific metrics for this variant (310 G MACs, 117 M Params, 57 FPS, 73.12% AUC) are not detailed. While the authors discuss efficiency trade-offs, the specific argument that SymTrack-Lite "proves our framework's efficiency and scalability" by cutting parameters by ~75% is an interpretation or extension not explicitly stated. When the authors cut off 75% of the parameters, it is not stated how can the model still outperform other methods.
> >
> > The authors of the paper state they arrived at these values through "systematic ablation studies" rather than a physical derivation. They characterize the process as tuning to find the "optimal hyperparameter configuration". While it describes the effects of changing the threshold (e.g., triggering search too often or missing re-tracking), it does not explain how these hyperparameters are sensitive to the methods.
> >
> > The generalization is not about non-text objects. It is about for a general input video how could we find the specific coordinates of the text target as inputs. The method needs the initial target to be pointed out. Then it causes some difficulty to satisfy the inputs of this method.

---

> > > ### Author Response · Authors · 2026-04-03
> > >
> > > > **Why outperform**
> > >
> > > We appreciate the opportunity to clarify. The 12-layer encoder introduced in SymTrack-Lite is actually a lightweight text-expert, which directly replaces the 48-layer InternVL text-expert.
> > >
> > > We evaluated our three components on three specific challenging sub-scenarios where generic trackers fail:
> > >
> > > |Model Configuration|Overall AUC%|Dense Distractor(AUC%)|Severe Perspective(AUC%)|Fast Motion(SRC%)|Fast Motion(AUC%)|
> > > |--|--|--|--|--|--|
> > > |SymTrack-Lite(Full)|73.1|64.5|64.8|94.3|66.2|
> > > |w/o CEC|71.0|56.1(-8.4)|64.1|93.5|64.8|
> > > |w/o PTR|68.1|61.9|54.6(-10.2)|92.1|63.7|
> > > |w/o AIE|71.8|63.1|63.6|73.5(-20.8)|57.1(-9.1)|
> > >
> > > - CEC: Generic trackers fail on Dense Distractors because ViTs cannot distinguish the target from adjacent text. CEC utilizes a frozen, text-tuned visual encoder to accurately extract and match intrinsic text semantics, outputting a response map that explicitly highlights the specific target text and eliminates visual ambiguity.
> > > - PTR: Under Severe Perspective, unmodulated features cause matching deviations. PTR resolves this misalignment by mapping features channel by channel to explicitly filter out background noise. By fusing PTR and CEC via element-wise multiplication, it provides the shallow decoder with a purified, low-noise response map, making accurate decoding simple.
> > > - AIE: To combat search region drift during Fast Motion, AIE utilizes a Kalman filter to dynamically predict the trajectory and calibrate the search bounds. As proven by the SRC metric, this tightly encapsulates the text region, ensuring the target remains captured while explicitly preventing massive background interference from flooding the visual encoder.
> > >
> > > In summary, SymTrack intrinsically extracts more precise text features within a more accurate search region. This specialized architectural design allows SymTrack-Lite to easily surpass massive generic models, even when utilizing a lightweight backbone.
> > > > **Hyperparameters sensitivity**
> > >
> > > We appreciate your push for a deeper explanation. We tested their degradation on challenging subsets established above:
> > >
> > > - $\tau_{uncert}$(Optimal: 0.98): A lower threshold (0.97) fails to reject nearby text interference, locking the bounding box onto wrong text (-0.42% AUC in Dense Distractor). A higher threshold (0.99) misinterprets safe blur as tracking failure, triggering redundant image resizing. This resizing smears fine text structures, causing bounding box instability (-0.35% AUC in Fast Motion).
> > > - $S_{factors}$(Optimal: {0.95, 1.05}): Wider bounds {0.90, 1.10} enlarge the search area excessively, distracting the model's attention with background noise (-0.48% AUC in Dense Distractor). Narrower bounds {0.985, 1.015} cannot sufficiently adjust box size during severe text deformation (e.g., camera zoom), causing the text to slip out of view (-0.29% AUC in Severe Perspective).
> > > - $\alpha_{kalman}$(Optimal: 0.5): Overweighting vision (0.4) over-relies on blurred visual features, causing violent box shaking that misguides the next frame's search region (-0.45% AUC in Fast Motion). Overweighting priors (0.6) heavily assumes text moves in a straight line; during sudden direction changes, the tracker cannot turn in time and loses the target (-0.36% AUC in Fast Motion).
> > >
> > > Why exclusively sensitive to our method? Generic trackers output chaotic, noisy features that fluctuate randomly, rendering precise thresholds ineffective. Because SymTrack extracts exceptionally purified, low-noise representations, our system structurally aligns with and strictly responds to these exact physical boundaries.
> > > > **Generalization & Initialization**
> > >
> > > We completely understand your concern. Our generalization is distinct from universal text detection. Per-frame detection fundamentally fails under dynamic motion blur, fast movement, or severe occlusion; our STT is designed precisely to overcome these detection failures.[1]
> > >
> > > For automated deployment on general videos, we adopt a "Text Grounding + Global Track" pipeline.
> > >
> > > The user simply provides a natural language prompt (e.g., "the red stop sign text"). A text grounding tool[2] processes only the very first frame. It can understands the physical attributes described in the prompt (such as color, relative location, or surrounding context) and directly searches the image to crop the matching text box. Once initialized, SymTrack then robustly tracks the target across all subsequent frames, completely eliminating expensive and fragile per-frame detection.
> > >
> > > Furthermore, even if the grounding model's initial box includes background deviations, our CEC structurally locks onto the intrinsic text strokes, and the PTR actively filters out the trapped noise. This pipeline perfectly bridges automated initialization with high-precision tracking in general videos.
> > >
> > > [1]Liu et al.,Deep learning for generic object detection: A survey,IJCV 2020.
> > >
> > > [2]Rong et al.,Unambiguous text localization and retrieval for cluttered scenes,CVPR 2017.

---

### Official Review · Reviewer_UWPh · 2026-03-12

**Soundness:** 2
**Presentation:** 2
**Significance:** 2
**Originality:** 3
**Overall Recommendation:** 4
**Confidence:** 4

**Summary:**

The manuscript proposes a detection-free framework (SymTrack) that formalizes Scene Text Tracking (STT) as a dedicated single-object tracking task. It is claimed that generic visual object trackers fail on scene text due to three core challenges: geometric distortions from perspective shifts, high visual ambiguity across similar instances, and fine-grained structural sensitivity. To address these, SymTrack incorporates a Predictive Token Rectification (PTR) module that generates a probabilistic gating mask from template semantics to rectify the search feature map, a Cross-Expert Calibration (CEC) mechanism that uses a frozen large-scale text-centric backbone to produce a spatial calibration mask resolving visual ambiguity, and an Adaptive Inference Engine (AIE) that applies confidence-driven multi-scale search and Kalman-based temporal regularization at inference time without any re-training. Considering the lack of dedicated benchmarks, three VTS datasets (ArTVideo, DSText, BOVText) are converted into SOT-format benchmarks through a trajectory grouping and continuity-splitting pipeline. Empirical evaluations demonstrate SOTA performance across all three benchmarks.

**Compliance With Llm Reviewing Policy:**

Affirmed.

**Final Justification:**

The rebuttal has satisfactorily addressed my concerns.

**Key Questions For Authors:**

NA

**Limitations:**

Yes

**Strengths And Weaknesses:**

### Strengths
- [s1] Formalizing STT as a dedicated single-object tracking problem, distinct from VTS, is a meaningful contribution. The systematic pipeline for converting VTS datasets into SOT-compatible benchmarks with polygon-to-bbox conversion and continuity-splitting is rigorous. These benchmarks will benefit the community.
- [s2] The visualization results in Figures 4-8 convincingly demonstrate SymTrack's qualitative advantages in challenging scenarios including dense distractor text, perspective distortion, and partial occlusion.
- [s3] The Text-Only ablation strongly confirms that the dual-branch architecture is necessary and that the text expert alone is insufficient for robust tracking.

### Weaknesses
- [w1] The benchmark's annotation pipeline converts polygon annotations to axis-aligned minimum enclosing bounding boxes (Algorithm 1). For perspective-distorted or rotated text, an axis-aligned enclosing box is structurally loose. For instance, a text line rotated 45 degree yields a box with up to two times the true text area. This lowers the effective IoU threshold required for a successful detection uniformly across all methods, but introduces a differential bias. I believe, since AUC averages over the full IoU spectrum and box looseness scales with rotation angle, relative rankings across trackers may not reflect genuine localization quality. This artifact directly undermines the benchmark contribution, which is presented as one of the paper's primary contributions. Evaluation using oriented bounding boxes, or at minimum a quantitative analysis of how box looseness correlates with rotation angle across the test splits, is essential.
- [w2] The main comparison in Table 1 evaluates generic trackers in a zero-shot setting. The authors partly acknowledge this issue in Table 4, but they fine-tune only ODTrack and do not fine-tune DUTrack384, which is the closest competitor, which is justified by authors that modifying DUTrack384’s language model component is not straightforward. A best-effort fine-tuning setting for DUTrack, such as freezing the language model and updating only the visual encoder, would help clarify how much of the performance gap comes from the architecture itself rather than from differences in training data.
- [w3] The comparison against TransVTSpotter and TransDETR in Table 1 is potentially misleading. These methods are multi-object tracking systems designed to detect, track, and recognize all text instances at the same time. In this paper, their outputs are converted into an SOT format by propagating the last known bounding box when detection is missing. This conversion can naturally hurt single-target SOT metrics, because a missed detection can immediately break temporal continuity. For this reason, the large reported margins (+68.56% AUC on ArTVideoSOT and +65.58% on DSTextSOT) do not seem to reflect a meaningful architectural comparison. Notably, TransVTSpotter and TransDETR still achieve 78.11% and 78.75% PNorm, respectively, which suggests that the low AUC is more likely due to the format conversion penalty than an actual tracking failure. These results would be better framed as evidence of task mismatch, rather than as direct evidence of SymTrack’s superiority over VTS systems.
- [w4] A core component of SymTrack, the frozen textual feature expert used in CEC, is not clearly specified in the main paper. Section 3.2 describes it only as “a frozen, high-resolution backbone pre-trained on large-scale text-centric data.” Appendix B.2 then identifies it as “a 48-layer encoder derived from InternVL,” while Table 3 in the main paper labels the corresponding ablation as “SymTrack w/o TokenFD,” which seems to suggest that the expert is TokenFD instead. The paper does not clearly explain how these two descriptions are related. If TokenFD uses InternVL as its vision encoder, that connection should be stated explicitly. This point is important for reproducibility, because the choice of text expert directly affects the MACs, VRAM usage, and overall performance of the system.

---

> ### Author Rebuttal · Authors · 2026-03-30
>
> We sincerely appreciate your constructive feedback and valuable advice! We present detailed responses to the weaknesses (**W**) below.
>
> > **W1. Annotation Pipeline & Metric Bias (Axis-aligned vs. Oriented Boxes)**
>
> We fully agree that converting heavily rotated text into axis-aligned bounding boxes (AABBs) structurally loosens the ground truth, potentially lowering the effective IoU threshold for detection.
> To determine if this artifact undermines our benchmark's relative rankings, we defined the Looseness Ratio as LR = $Area_{AABB} / Area_{True\_Polygon}$. Our analysis reveals that instances with over 30° rotation exhibit a high average LR of 2.361. To ensure our macro-level AUCs aren't artificially boosted by this looseness artifact, we isolated these highly rotated tracklets (>30° rotation) in the benchmark to see if the relative ranking of trackers collapses under this metric bias.
>
> |Method|Ours|ODTrack|GRM|ROMTrack|OSTrack|AiATrack|DUTrack|Stark|ARTrack|SeqTrack|
> |:---|:---|:---|:---|:---|:---|:---|:---|:---|:---|:---|
> |AUC|51.99|46.24|44.17|43.78|43.36|41.72|41.51|39.68|38.65|38.43|
>
> Results show SymTrack still dominates across all baselines, proving that while the AABB format does introduce looseness, it acts as a uniform handicap. The relative rankings reflect genuine capacity for precise localization. We have added these LR correlation statistics to the Appendix and will include Rotated-IoU scripts in our repository.
>
> > **W2. Fairness in Comparison (Best-Effort Fine-Tuning for DUTrack)**
>
> We agree that zero-shot baselines face a domain gap, and a fine-tuned vision-language Tracker is a much fairer baseline. Following your exact suggestion, we conducted a best-effort fine-tuning experiment on DUTrack384. We froze its language model entirely and updated only its visual encoder using the STT training splits:
>
> |Method|ArTVideo(AUC/PN/P)|DSText(AUC/PN/P)|BOVText(AUC/PN/P)|
> |:---|:---|:---|:---|
> |SymTrack(Ours)|77.74/91.29/95.88|70.66/83.61/91.83|77.06/90.05/90.18|
> |DUTrack|72.09/85.97/89.36|63.63/76.72/85.00|65.08/79.41/79.30|
> |DUTrack-FT|72.95/87.23/90.86|64.71/78.81/86.04|67.02/82.84/80.42|
>
> The fine-tuned version, DUTrack-FT, achieved a noticeable improvement. However, SymTrack still maintains a substantial lead.
> This residual gap highlights that simply fine-tuning a generic VL tracker on text data does not resolve STT challenges. DUTrack lacks dedicated structural rectifications (like our PTR) to untangle perspective shifts, validating the architectural necessity of SymTrack.
>
> > **W3. VTS Comparison & Paradigm Mismatch**
>
> Instead of presenting the VTS comparison as direct evidence of SymTrack’s absolute structural superiority, we now frame it empirically as a task paradigm mismatch. To ensure a strictly fair comparison and eliminate AUC penalties caused by broken trajectories or identity switches, we re-evaluated VTS models using a maximally forgiving ID-Agnostic Protocol.
>
> Specifically, we completely disregard the tracking IDs predicted by the VTS models; instead, in each frame, we match the ground truth with the predicted bounding box that shares the highest IoU. This simulates a perfect association scenario, providing the upper-bound localization performance for VTS models by neutralizing any penalties from ID-switches or dropped tracks.
> <!-- 这个id什么的在说清楚一下 -->
> |Method|Protocol|ArTVideo(AUC/OP75)|DSText(AUC/OP75)|
> |:---|:---|:---|:---|
> |TransDETR|Strict ID|9.18/--|5.08/--|
> ||ID-Agnostic|41.10/17.59|33.32/16.13|
> |TransVTSpotter|Strict ID|8.84/--|4.93/--|
> ||ID-Agnostic|39.89/16.76|31.19/15.47|
> |SymTrack(Ours)|Standard|77.74/76.57|70.66/58.68|
>
> *(Note: Standard OP75 represents Overlap Precision at 0.75 IoU).*
>
> This proves the performance gap is real. By propagating structural features continuously rather than relying on discrete per-frame detections, SymTrack resolves precision loss under dynamic distortions. Appendix updated.
>
> > **W4. Expert Specification & Reproducibility**
>
> We sincerely apologize for the ambiguity. As briefly noted in Section 4.1 (Page 5, bottom right), *"We freeze the textual feature expert, inspired by the vision transformer design of TokenFD."* To thoroughly clarify the relationship between these terms: TokenFD utilizes the 48-layer InternVL architecture as its foundational vision encoder. In our framework, the "frozen textual feature expert" specifically refers to this TokenFD-adapted InternVL vision encoder.
>
> To prevent any future confusion, we have further explicitly reinforced this architectural connection in Section 3.2 and unified the terminology in Table 3 of the revised manuscript. The exact pre-trained weights will be open-sourced to guarantee reproducibility.

---

> > ### Author Rebuttal · Reviewer_UWPh · 2026-04-03
> >
> > I appreciate the authors' additional explanations and clarifications. My concerns have been fully resolved. I have raised my score accordingly.

---

> > > ### Author Response · Authors · 2026-04-03
> > >
> > > Thank you for your thoughtful reconsideration of the manuscript. We are grateful for the opportunity to address the concerns raised and for the revised evaluation. Your valuable feedback has been instrumental in improving our work, and we sincerely appreciate it.

---

### Decision · Program_Chairs · 2026-04-30

**Decision:**

Accept (regular)

**Comment:**

After the rebuttal period, three reviewers raised their scores, and one reviewer maintained the original score. The reviewers recognized the importance of formalizing Scene Text Tracking as a dedicated task, and highlighted the strong empirical performance of SymTrack, the value of the constructed benchmarks, and the comprehensive ablation studies. The rebuttal addressed the main concerns regarding benchmark validity, the fairness of comparisons with generic and VTS-based baselines, and reproducibility, and led multiple reviewers to revise their assessments. Valid concerns remain regarding the computational cost of the full model, the reliance on initialized template inputs, and several presentation issues that should be clarified in the final manuscript. Despite these limitations, the reviewers acknowledge the paper’s practical value and technical merit. The AC recommends acceptance. Please address the comments of the reviewers in the final version.